# Sound-meaning associations allow listeners to infer the meaning of foreign language words

Sayuri Hayakawa [1]✉ & Viorica Marian [2]

An attribute of human language is the seemingly arbitrary association between a word's form and meaning. We provide evidence that the meaning of foreign words can be partially deduced from phonological form. Monolingual English speakers listened to 45 antonym word pairs in nine foreign languages and judged which English words corresponded to these words' respective meanings. Despite no proficiency in the foreign language tested, participants' accuracy was higher than chance in each language. Words that shared meaning across languages were more likely to share phonological form. Accuracy in judging meaning from form was associated with participants' verbal working memory and with how consistently phonological and semantic features of words covaried across unrelated languages. A follow-up study with native Spanish speakers replicated the results. We conclude that sound maps to meaning in natural languages with some regularity, and sensitivity to form-meaning mappings indexes broader cognitive functions.

[1] Oklahoma State University, Stillwater, OK, USA. [2] Northwestern University, Evanston, IL, USA. ✉email: sayuri.hayakawa@okstate.edu

Language is a symbolic system, one in which conceptual referents are assigned to sound, letter, or sign symbols[1]. Even if the meaning of a word (e.g., "mountain") is transparent to a speaker of the language, the semantic and perceptual qualities of the referent (e.g., the characteristics of a mountain) are not contained in the wordform itself. Yet people often share remarkably similar intuitions about the meaning of a word they have not encountered before. When shown pictures of novel objects (e.g., a spikey object with sharp edges and a cloud-like object with rounded edges) and asked to identify which one is a "kiki" and which one is a "bouba," the label "kiki" is consistently assigned to objects with sharp edges while "bouba" is more likely to be assigned to round objects[2,3]. This propensity to draw connections between sensorimotor qualities of speech and the external world may underly the evolution of language itself[4–6]. By presenting participants with words and concepts that do not exist in any language (and thus cannot be matched based on learned conventions), early research provides compelling evidence that people form systematic associations between the sounds and meanings of words (i.e., "sound symbolism"). Although listeners can also cue into form-meaning regularities to correctly identify the referents of existing foreign words[7–14], the cognitive processes that enable the extraction of meaning from sound in natural languages are less understood. The goal of the present study is to investigate if, and when, listeners can infer the meanings of foreign words based on their phonological forms alone, with a particular focus on whether individual differences predict the ability to deduce the meanings of non-native wordforms. The present study examines whether cross-linguistic regularity in form-meaning associations and behavioral measures of cognitive abilities predict the extraction of meaning from foreign language wordforms in nine different languages.

The capacity to correctly identify the meanings of unfamiliar wordforms depends not only on the presence of non-arbitrary associations between concepts and labels, but also on the ability to capitalize on form-meaning regularities within and across languages. Existing research on sound symbolism is primarily concerned with the objective or perceived resemblance between the acoustic features of words (e.g., vowel height and backing, consonant voicing) and attributes of their referents. The resemblance between form and meaning is more broadly captured by the term *linguistic iconicity*, and can emerge with respect to the sound of a word (as in sound symbolism), as well as other perceptual and articulatory features (e.g., between the visual appearance of a butterfly and the fluttering hand gesture associated with its label in American Sign Language). Furthermore, while linguistic iconicity is most readily apparent when words and referents share qualities within a given modality (e.g., onomatopoeias such as "beep" that directly mimic sounds of the referent), analogical associations can also emerge across modalities. For instance, a word with longer and more open vowels (e.g., /a/) may be more readily associated with larger objects than shorter vowels (e.g., /i/), and consonants produced by a flow of air (e.g., /m/) may be more intuitively linked to smooth objects than those produced by a sudden air blockage (e.g., /k/). Cross-modal associations have been observed between the sounds of vowels and consonants and object features such as shape (e.g., voiceless consonants with spikey, voiced consonants with round)[2,3,15–19], size (e.g., /i/ associated with small, /a/ associated with big)[20–24], lightness (e.g., voiceless consonants with light, voiced consonants with dark)[25], touch (e.g., voiced consonants with rough, voiceless consonants with smooth)[26], and taste (e.g., front-vowels with sour, back-vowels with sweet)[27–30], as well as with semantic characteristics such as level of abstractness (e.g., front-vowels with concrete, back-vowels with abstract)[31] and even personality (e.g., sonorant consonants with conscientiousness, plosive consonants with extraversion)[32,33].

Form-meaning connections have been explored in speakers of different languages[3,18,21,34] and at different developmental stages[16,19]. However, because investigations typically make use of carefully manipulated non-words that enable strict control over phonological features of stimuli[3,7,16–19,23,31,34–40], the observed associations between sound and meaning have been questioned under the assumption that iconicity is rare in natural languages. Although the use of non-words is invaluable for isolating particular form-meaning associations and controlling for prior language experience, phonological features of artificial stimuli are often designed to maximally differ from one another and may fail to generalize to natural languages[41]. The mechanisms and moderators of sound-meaning mapping have been relatively understudied using natural language stimuli. When used, natural language words are frequently restricted to ideophones such as onomatopoeia (most often in Japanese due to the high number of ideophone words), which limits our understanding of sound symbolism during language processing more generally. A number of recent studies utilizing large cross-linguistic datasets indicate that systematic form-meaning associations can indeed be found in natural languages[9,42]. For instance, Winter et al.[42] demonstrated that words related to sensations of roughness are particularly likely to include a trilled /r/ sound, and that this pattern holds true across a diverse set of languages and language families. To more closely examine the role of iconicity in human language, it is critical to not only uncover systematic links between form and meaning in natural languages, but to ground the phenomenon within the broader cognitive system.

Sound-meaning mapping may be supported by more general mechanisms underlying cross-modal integration. For instance, when presented with kiki-bouba type tasks, individuals with developmental disorders impacting cross-modal integration (e.g., dyslexia[36]; autism spectrum disorder[37]) are less consistent in how they map novel words to meanings relative to controls. At the other end of the continuum, synesthetes (who experience extreme forms of cross-modal integration such as seeing colors when hearing words) more accurately guess the meaning of foreign words than non-synesthetes[7]. Neural correlates of form-meaning mapping suggest that the ability to maintain and bind cross-modal information may play a key role in establishing perceptual links between words and their referents[8,38,43–45]. Revill et al.[8] found increased activation of the left superior parietal cortex for sound symbolic vs. non-symbolic words, and sensitivity to sound-meaning regularities was predicted by individual differences in functional anisotropy in the left superior longitudinal fasciculus (SLF). The SLF, also referred to as the arcuate fasciculus (AF), connects frontal, temporal, and parietal language areas to one another and is associated with a number of language-related functions including cross-modal integration, binding articulatory and acoustic information, and verbal working memory[46–48]. Kanero et al.[44] observed increased activation of the right superior temporal sulcus for sound symbolic compared to non-symbolic words, and posit that the region may support cross-modal integration of acoustic word features and perceptual qualities of their referents.

Connecting behavioral and neuroanatomical differences, Margiotoudi et al.[45] found evidence of sound symbolism in humans, but not in great apes. They proposed that form-meaning connections may stem in part from humans' richer development of the AF and that the relative deficit in form-meaning mapping among non-human primates may result from their limited ability to establish and maintain associations between phonological and semantic representations in verbal working memory. Form-meaning mapping may additionally be supported by mirror neurons in regions associated with verbal working memory and phonetic rehearsal (e.g., the inferior frontal gyrus) that enable the

representation of natural acoustic events through articulatory imitation[38].

To determine whether systematic form-meaning regularities are present in natural languages, we examined whether foreign language words with the same meaning share more phonological features across languages than words with different meanings. To understand the linguistic and cognitive mechanisms behind the extraction of meaning from form, we examined (1) whether the ability to infer meaning from form increases with the degree to which a given word conforms to cross-linguistic regularities in form-meaning mapping, and (2) whether detection and use of form-meaning regularities are moderated by individual differences in verbal working memory. Based on prior evidence linking working memory, word recognition, and the maintenance of cross-modal representations[8,45,47,49,50], we predict that better verbal working memory will enhance sensitivity to form-meaning regularities, leading to more accurate semantic decoding of word forms .

We tested these predictions using a forced-choice antonym task developed by Tsuru and Fries[10] and modified by D'Anselmo et al.[12]. Native monolingual English speakers listened to pairs of non-ideophonic foreign words (e.g., "nibui [鈍い]… surudoi [鋭い]", the Japanese words for "blunt" and "sharp") and identified which of two English word pairs corresponded to their respective meanings (e.g., blunt : sharp or sharp : blunt). To establish the generalizability of the phenomenon, the study included equal numbers of nouns, verbs, and adjectives across nine foreign languages with variable degrees of similarity to English (a Germanic Indo-European language). These included Japanese, Mandarin, and Thai (belonging to the Japonic, Sino-Tibetan, and Tai-Kadai language families, respectively; henceforth referred to as Japonic-Sino-Tai languages), Polish, Russian, and Ukrainian (Slavic Indo-European languages), and French, Romanian, and Spanish (Romance Indo-European languages). Phonetic distances between words were used to examine how accuracy varied depending on similarity to the native language (English), as well as to test the prediction that foreign words with the same meaning would have greater phonological overlap across languages than words with different meanings. Individual differences in verbal working memory were assessed with an auditory digit span task and used to examine the relationship between verbal working memory capacity and sound-meaning mapping during natural language processing. To determine the generalizability of effects to a different language population, a replication study utilized the same procedure and stimuli with native Spanish speakers tested in a Japonic-Sino-Tai language (Japanese), a Slavic language (Polish), and a Germanic language (English).

## Methods

**Participants**. Participants were recruited online using the Prolific platform (www.prolific.co) and the survey was completed using Qualtrics (https://www.qualtrics.com). Informed consent was obtained in accordance with the protocol approved by Northwestern University's Institutional Review Board. The reported studies were not preregistered. An a priori power analysis for logistic regression using G*Power 3.1.9.7[51] with an alpha level of 0.05 and estimated power of 0.80 indicated a necessary sample size of 98 to detect an odds ratio of 1.86 (H0 = 0.5 and H1 = 0.65 based on pilot data).

*Native English speakers*. To account for variability in the online population, a total of 134 monolingual English speakers with a mean age of 35.61 years old (SD = 11.60) were included in the first study. Forty-six percent of participants self-reported their gender as female and 54% reported their gender as male. The

majority of participants self-identified as non-Hispanic White (77.6%), followed by Black or African American (13.4%), More than One Race (5.2%), and Asian (2.3%). An additional two participants (1.5%) declined to answer. Two additional participants were excluded due to technical issues and 18 additional participants dropped out following eligibility screening. Volunteers were screened for eligibility prior to participation. Inclusionary criteria included native language status and proficiency in English (8 or higher on a 0–10 scale; M = 9.87, SD = 0.34), with minimal proficiency in each of the nine foreign languages used in the experiment (0 or 1 on a 0–10 scale for Japanese, Mandarin, Thai, Polish, Russian, Ukrainian, French, Romanian, Spanish; M = 0.05, SD = 0.11 across languages). Each eligible participant completed the antonym word pair task in three foreign languages (one Japonic-Sino-Tai, one Slavic, and one Romance language), resulting in 45 participants each for Japanese, Mandarin, Polish, Ukrainian, Romanian, and Spanish and 44 participants each for Thai, Russian, and French. Data were collected between 02/22/ 2022-03/18/2022 and participants were compensated monetarily.

*Native Spanish speakers*. The follow-up replication study included a total of 46 monolingual Spanish speakers with a mean age of 28.63 years old (SD = 7.03). Thirty-seven percent of participants self-reported their gender as female and 63% reported their gender as male. The majority of participants self-identified as Hispanic (78.3%), followed by non-Hispanic White (10.9%), More than One Race (8.7%), and Native American (2.1%). One additional participant was excluded due to perfect performance on the English word-pair task despite reporting minimal English experience and no participants dropped out following eligibility screening. Inclusionary criteria included native language status and proficiency in Spanish (8 or higher on a 0–10 scale; M = 9.82, SD = 0.49), with minimal proficiency in each of the three foreign languages used in the experiment (0 or 1 on a 0–10 scale for Japanese, M = 0.22, SD = 0.42, and Polish, M = 0.11, SD = 0.31; and 0–3 for English, M = 1.48, SD = 0.81). A higher threshold was set for English due to the relatively low number of Spanish speakers with no exposure to English. Each eligible participant completed the antonym word pair task in three foreign languages: Japanese, the Japonic-Sino-Tai language with the median effect size in the initial study; Polish, the Slavic language with the median effect size in the initial study; English, the Germanic native language from the initial study. Romance languages were not included in the replication due to substantial overlap with the native language of Spanish. Data were collected between 08/01/ 2023 and 08/13/2023 and participants were compensated monetarily.

**Stimuli**. We began by gathering an initial set of 60 antonym pairs from lists of common English words. Native speakers of the non-English languages were then consulted to remove word pairs with ambiguous, inexact, or otherwise problematic translations and select a final set of 45 English antonym word pairs. These included 15 noun pairs (mountain-valley), 15 verb pairs (sink-float), and 15 adjective pairs (dry-wet; see Table 1). The pairs were matched for English lexical frequency[52], concreteness[53], and valence, arousal, and dominance[54] across the three parts of speech (all p > 0.05). For the non-English stimuli, nine languages comprised of three Japonic-Sino-Tai languages (Japanese, Mandarin, Thai), three Slavic Indo-European languages (Polish, Russian, Ukrainian), and three Romance Indo-European languages (French, Romanian, Spanish) were chosen to represent a range of language families with variable degrees of similarity to English. The English word pairs were translated into each of the nine foreign languages by native speakers. Two digitized text-to-

| Table 1 English antonym word-pair stimuli. | | | | | |
| --- | --- | --- | --- | --- | --- |
| **Adjectives** | | **Nouns** | | **Verbs** | |
| **Word1** | **Word2** | **Word1** | **Word2** | **Word1** | **Word2** |
| Dry | Wet | Mountain | Valley | Sink | Float |
| Hot | Cold | Boy | Girl | Yell | Whisper |
| Thick | Thin | Child | Adult | Push | Pull |
| Black | White | Entrance | Exit | Rise | Fall |
| Alive | Dead | North | South | Melt | Freeze |
| Asleep | Awake | Morning | Evening | Write | Read |
| Bitter | Sweet | Hero | Villain | Buy | Sell |
| Blunt | Sharp | Answer | Question | Close | Open |
| Small | Big | Ally | Enemy | Attack | Defend |
| Shallow | Deep | Peace | War | Hide | Reveal |
| Slow | Fast | Poverty | Wealth | Come | Go |
| Loud | Quiet | Beginning | Ending | Accept | Decline |
| Sick | Healthy | Failure | Success | Create | Destroy |
| Wide | Narrow | Heaven | Hell | Exclude | Include |
| Happy | Sad | Pride | Shame | Forget | Remember |

voice services (www.play.ht and www.ttsmp3.com) were used to construct the auditory stimuli. Each auditory foreign language word pair was spoken by a female voice with a 1 s pause between words. Auditory stimuli were amplitude-normalized and checked by native speakers to ensure correct pronunciations. All stimuli, including foreign language translations, IPA transcriptions, and audio recordings are available at https://osf.io/4ez8v/.

**Procedure**. Following screening and informed consent, eligible participants were randomly assigned to complete the antonym word pair task in three languages (one Japonic-Sino-Tai language, one Slavic language, and either one Romance language for native English speakers or English, a Germanic language for native Spanish speakers) in three separate blocks. Language assignments and block order were counterbalanced across participants and trial order was randomized within blocks. On each trial, participants listened to a pair of sequentially spoken foreign words (e.g., "*nibui* [鈍い]... *surudoi* [鋭い]") and identified which of two configurations of written native language translations corresponded to their meanings (e.g., sharp : blunt or blunt : sharp; see Fig. 1).

The sequential order of spoken words within each pair was counterbalanced across participants (e.g., "*nibui* [鈍い]... *surudoi* [鋭い]" or "*surudoi* [鋭い]... *nibui* [鈍い]") and the right vs. left position of the correct native language translation on the screen was randomized on each trial. The native language choice options appeared on screen at trial onset and participants clicked a button to play the audio a minimum of one time and a maximum of two times before responding. Although no time limit was enforced, participants were asked to respond within 20 s and were shown a 20 s count down timer on the screen. To enhance engagement, participants were provided with their accuracy score at the end of each language block. Participants completed two practice trials at the beginning of the experiment, followed by 45 trials in each of the three blocks for a total of 135 experimental trials.

Following the antonym task, participants were shown the native language translations of all words heard during the experiment and were asked to indicate whether any of their foreign language equivalents were recognized or known in each of the assigned languages. Trials which included a previously known foreign language translation were discarded on a participant-by-participant basis (4.44% of all trials in the native English speaker study; 14.70% of all trials in the native Spanish speaker replication study). Lastly, participants completed a verbal working memory

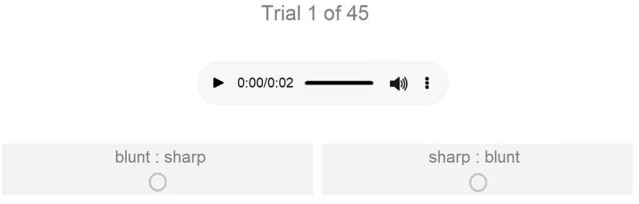

Trial 1 of 45

| blunt : sharp | sharp : blunt |

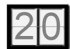

**Fig. 1 Example response display for the antonym word task with native English speakers.** Participants saw an English word pair in two configurations (e.g., blunt : sharp and sharp : blunt) and clicked a button to listen to a pair of sequentially spoken foreign translations (e.g., "*nibui* [鈍い]... *surudoi* [鋭い]", meaning "blunt" and "sharp" respectively). Participants indicated which of the English word pairs corresponded to the foreign words' meanings. The same procedure and stimuli were used in the replication with all written English word pairs replaced with Spanish word pairs.

test (digit span subtest of the *Comprehensive Test of Phonological Processing*[55]) and a demographic and language background survey (*Language Experience and Proficiency Questionnaire*[56]; see Fig. 2).

**Data analysis**. Accuracy for each word pair was coded as a binomial outcome variable (0 = incorrect, 1 = correct). Effects of Language (Japanese, Mandarin, Thai, Polish, Russian, Ukrainian, French, Romanian, Spanish for native English speakers; Japanese, Polish, English for native Spanish speakers in the replication study), Part of Speech (Noun, Verb, Adjective), Phonetic Distance from the native language, Form-Meaning Regularity, and Verbal Working Memory on accuracy were analyzed using generalized linear mixed effects models with the *glmer* function of the *lme4* package in R[57]. The initial model included fixed effects of Language, Part of Speech, and their interaction, with both predictors sum coded to compare each level to the grand mean. The maximally converging model additionally included random intercepts for participant and item. Significance of fixed effects was assessed with chi-square tests and individual parameters were estimated with the Satterthwite method using the *lmerTest* R package[58]. Effect sizes for the generalized linear mixed effects models were estimated with odds ratios and all tests were two-sided. Follow-up tests of simple effects were conducted using the *emmeans* and *emtrends* functions of the *emmeans* R package[59]. Overdispersion and homogeneity of variance was checked using the *DHARMa* R package[60].

Two additional analyses examined effects of (1) phonetic distance from the native language (English or Spanish) and (2) verbal working memory and degree of form-meaning regularity. Phonetic distance from the native language was operationalized by first phonetically transcribing each native and foreign word into the International Phonetic Alphabet (IPA) and computing the phonetic distance between the phonological form of each foreign word and its corresponding native translation. Phonetic distance was calculated using the Sounded Cognates Algorithm (SCA) method implemented through the LingPy Python library[61]. SCA is a sound-class-based approach whereby phonetic units are first assigned to broader phonetic classes (e.g., stops, fricatives, vowels). The similarity between pairs of words are then determined based on their phonetic class memberships, with higher values indicating greater phonetic distance. The effect of phonetic distance on accuracy

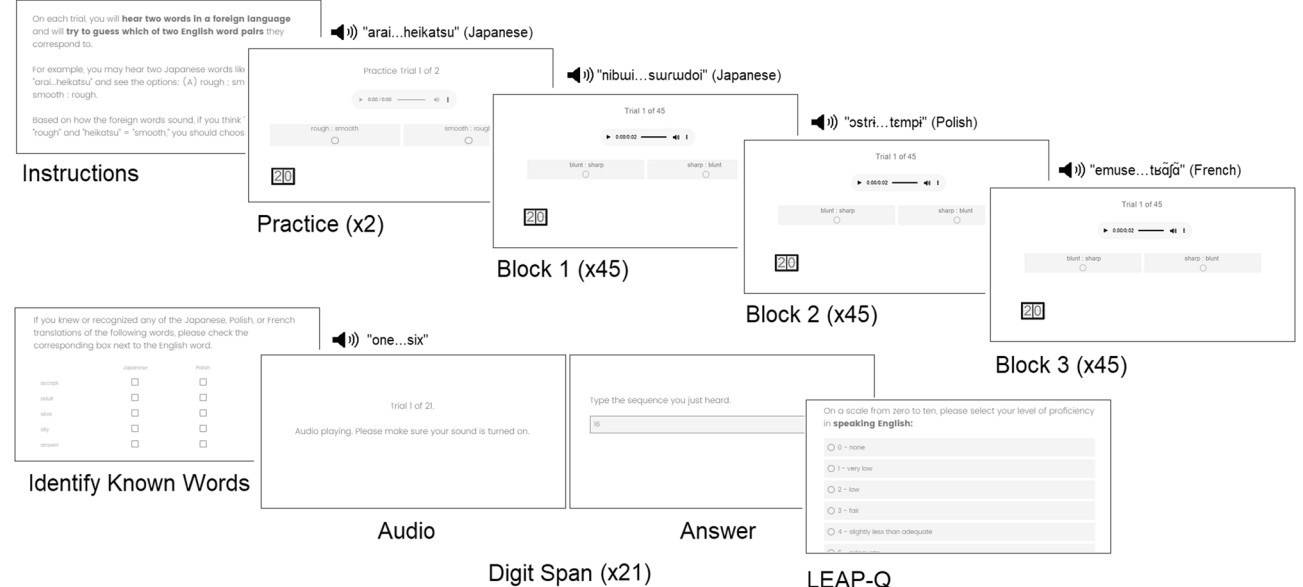

**Fig. 2 Overview of study procedure.** Following instructions for the antonym word pair task, participants completed two practice trials followed by three blocks of 45 trials each in a Japonic-Sino-Tai, Slavic, or Romance language for native English speakers and a Japonic-Sino-Tai (Japanese), Slavic (Polish), or Germanic (English) language in the native Spanish replication. Participants then identified any foreign language words they were already familiar with. This was followed by 21 trials of a digit span task during which participants heard a string of numbers in English (initial study) or Spanish (replication study) and then typed the number they heard from memory. The experiment concluded with the LEAP-Q to obtain detailed demographic and language background information.

was assessed with a generalized linear mixed effects model with sum coded fixed effects of Language Group (Japonic-Sino-Tai, Slavic, Romance or Germanic) and Part of Speech (Noun, Verb, Adjective), z-score transformed phonetic distance from the native language, and all interactions. The maximally converging model additionally included random intercepts for participant and item.

To obtain a measure of each word pair's degree of form-meaning regularity, we began by calculating the average phonetic distance between each word and its translation in the foreign languages outside of its language group (e.g., the average phonetic distance between the Japanese word *nibui* [鈍い], meaning "blunt," and its translation equivalents in Polish, Russian, Ukrainian, French, Romanian, and Spanish). We then subtracted this value from the average distance between the target word and the foreign language translations of its antonym (e.g., between *nibui* [鈍い] and the translation equivalents of *surudoi* [鋭い], meaning "sharp"). Positive form-meaning regularity scores therefore represented the extent to which words with the same meaning shared more phonological features across unrelated languages than words with opposite meanings. Effects of form-meaning regularity and verbal working memory were assessed using a generalized linear mixed effects model with accuracy as the outcome variable and fixed effects of Language Group, Part of Speech, z-score transformed form-meaning regularity score, z-score transformed digit span score, and all two-and three-way interactions. The maximally-converging model also included random intercepts for participant and item. Participants with digit span scores 1.5 standard deviations or more below the mean ($N = 12$ native English speakers; $N = 3$ native Spanish speakers) were excluded from the analysis.

**Reporting summary**. Further information on research design is available in the Nature Portfolio Reporting Summary linked to this article.

## Results

**Native English speakers: effects of language and part of speech.** Levene tests indicated no evidence of heterogeneity of variances across languages ($F(8) = 1.10$, $p = 0.356$) or parts of speech ($F(2) = 0.54$, $p = 0.581$). A simulation-based dispersion test of residual variances indicated no evidence of overdispersion ($\chi^2 = 1.0003$, $p = 0.960$), suggesting that the observed variability in the data was not significantly greater than what would be expected based on the model structure. A significant intercept term denoting deviation from 50% accuracy indicated that, overall, participants were able to correctly identify the meanings of foreign language antonym pairs better than what would be expected by chance ($z = 11.38$, $p < 0.001$, Estimate $= 0.60$, SE $= 0.05$, 95% CI[0.50, 0.70]). Sidak-adjusted follow-up tests revealed above-chance accuracy for each of the nine languages, including Japanese ($z = 3.04$, $p = 0.021$, Estimate $= 0.55$, SE $= 0.02$, 95% CI[0.52, 0.59]), Mandarin ($z = 2.86$, $p = 0.037$, Estimate $= 0.55$, SE $= 0.02$, 95% CI[0.52, 0.58]), Thai ($z = 4.25$, $p < 0.001$, Estimate $= 0.57$, SE $= 0.02$, 95% CI[0.54, 0.61]), Polish ($z = 4.38$, $p < 0.001$, Estimate $= 0.58$, SE $= 0.02$, 95% CI[0.54, 0.61]), Russian ($z = 3.41$, $p = 0.006$, Estimate $= 0.56$, SE $= 0.02$, 95% CI[0.53, 0.59]), Ukrainian ($z = 4.84$, $p < 0.001$, Estimate $= 0.58$, SE $= 0.02$, 95% CI[0.55, 0.62]), French ($z = 16.36$, $p < 0.0001$, Estimate $= 0.79$, SE $= 0.01$, 95% CI[0.76, 0.82]), Romanian ($z = 14.05$, $p < 0.0001$, Estimate $= 0.74$, SE $= 0.01$, 95% CI[0.71, 0.77]), and Spanish ($z = 17.58$, $p < 0.0001$, Estimate $= 0.81$, SE $= 0.01$, 95% CI[0.78, 0.83]; see Fig. 3).

There was a significant main effect of Language ($\chi^2(8) = 668.27$, $p < 0.001$), as well as a significant interaction with Part of Speech ($\chi^2(16) = 134.59$, $p < 0.001$). Tukey-adjusted pairwise comparisons revealed that accuracy was significantly higher for each of the three Romance Indo-European languages (French, Romanian, Spanish) than for each of the Slavic Indo-European languages (Polish, Russian, Ukrainian) and Japonic-Sino-Tai languages (Japanese, Mandarin, Thai; all $p < 0.001$), and higher for French ($z = 3.44$, $p = 0.017$, odds ratio $= 1.34$, 95% CI[1.03, 1.75]) and Spanish ($z = -4.66$, $p < 0.001$, OR $= 0.67$,

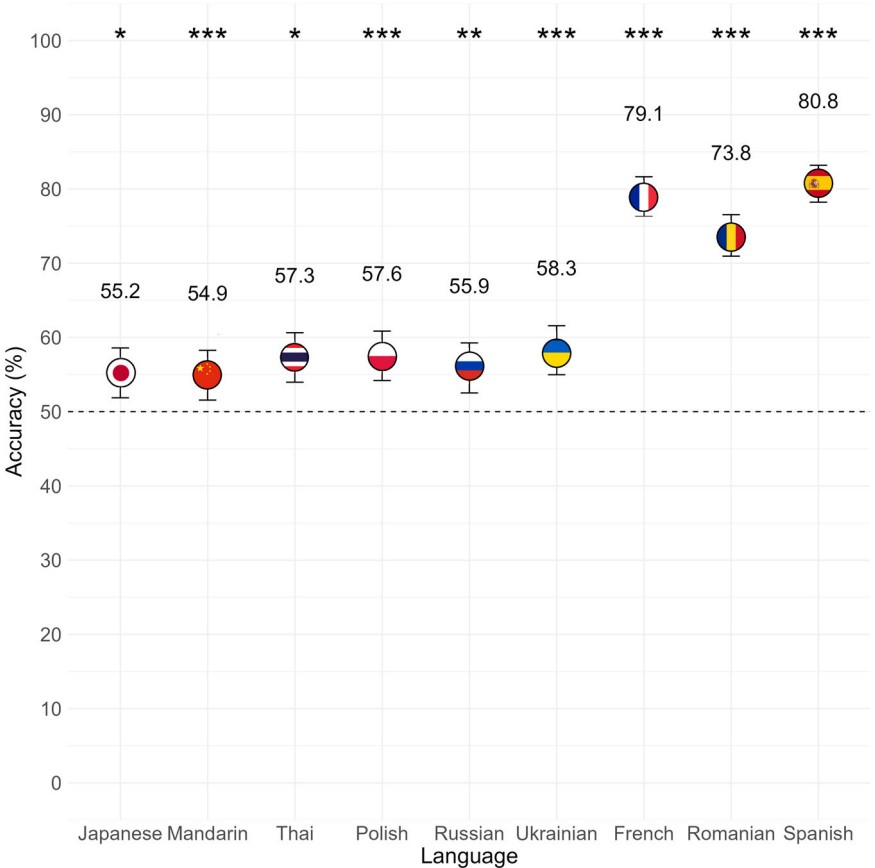

**Fig. 3 Mean accuracy by language for native English speakers.** Participants correctly identified the meaning of antonym word pairs significantly better than chance (dashed horizontal line) in each of the nine foreign languages (*n* = 45 independent participants each for Japanese, Mandarin, Polish, Ukrainian, Romanian, Spanish; *n* = 44 independent participants each for Thai, Russian, French). Error bars represent 95% confidence intervals. *p < 0.05, **p < 0.01, ***p < 0.001.

95% CI[0.51, 0.87]) compared to Romanian. We found no evidence that accuracy differed between or among any of the Slavic or Japonic-Sino-Tai languages; all *p* > 0.05; see Table 2).

Pairwise comparisons of parts of speech within each language revealed significantly higher accuracy for nouns relative to adjectives in French (*z* = −8.27, *p* < 0.001, OR = 0.20, 95% CI[0.12, 0.31]) and Romanian (*z* = −2.66, *p* = 0.023, OR = 0.63, 95% CI[0.42, 0.95]), and marginally higher accuracy for nouns relative to adjectives in Spanish (*z* = −2.25, *p* = 0.072, OR = 0.64, 95% CI[0.40, 1.02]). For French, accuracy was additionally higher for nouns relative to verbs (*z* = 3.59, *p* = 0.001, OR = 2.09, 95% CI[1.29, 3.38]), and for verbs relative to adjectives (*z* = −5.07, *p* < 0.001, OR = 0.41, 95% CI[0.27, 0.62]; see Fig. 4). We found no evidence that accuracy differed by part of speech in the other languages (all *p* > 0.05; see Table 3).

**Effects of language group and phonetic distance from English.** Follow-up analyses included a continuous measure of phonetic distance from English, capturing the extent of phonological overlap between each of the English and foreign language word pairs (with higher values indicating greater phonetic distance). Phonetic distance from English significantly differed across language groups ($\chi^2(2) = 57.01$, *p* < 0.001), with shorter distances for Romance Indo-European languages (*M* = 0.67, SD = 0.20, 95% CI[0.64, 0.69]) than for both Slavic Indo-European (*M* = 0.74, SD = 0.12, 95% CI[0.71, 0.77]; *t*(354) = 4.69, *p* < 0.001, Estimate = 0.08, SE = 0.02, 95% CI[0.04, 0.11]) and Japonic-Sino-Tai languages (*M* = 0.79, SD = 0.12, 95% CI[0.76, 0.82]; *t*(354) = 7.47, *p* < 0.001, Estimate = 0.12, SE = 0.02, 95% CI[0.08,

0.16]). Average distance from English was additionally shorter for Slavic languages than Japonic-Sino-Tai languages (*t*(354) = 2.78, *p* = 0.016, Estimate = 0.05, SE = 0.02, 95% CI[0.01, 0.08]). A significant interaction between Language Group and Part of Speech ($\chi^2(4) = 46.98$, *p* < 0.001) revealed that for Romance languages, average distance was significantly shorter for nouns than for verbs (*t*(115) = −3.35, *p* = 0.003, Estimate = −0.12, SE = 0.04, 95% CI[−0.20, −0.03]) and adjectives (*t*(115) = 5.96, *p* < 0.001, Estimate = 0.21, SE = 0.04, 95% CI[0.13, 0.30]), and shorter for verbs than adjectives (*t*(115) = 2.62, *p* = 0.027, Estimate = 0.09, SE = 0.04, 95% CI[0.01, 0.18]). We found no evidence that average distance varied by part of speech for Slavic or Japonic-Sino-Tai languages (see Table S1).

Model tests indicated no evidence of heterogeneity of variances across languages (*F*(2) = 0.69, *p* = 0.503) or parts of speech (*F*(2) = 0.94, *p* = 0.389), and no evidence of overdispersion ($\chi^2 = 1.01$, *p* = 0.576). Accuracy increased with shorter phonetic distances from English ($\chi^2(1) = 178.98$, *p* < 0.001, OR = 0.41, 95% CI[0.68, 0.75]), which was qualified by a significant three-way interaction between Language Group, Part of Speech, and Phonetic Distance ($\chi^2(4) = 75.47$, *p* < 0.001). For Romance languages, Sidak-adjusted tests of simple effects revealed that accuracy significantly increased with shorter phonetic distances for nouns (*z* = −14.30, *p* < 0.001, OR = 0.35, 95% CI[0.31, 0.41]) and verbs (*z* = −10.72, *p* < 0.001, OR = 0.40, 95% CI[0.34, 0.48]), but we found no evidence of a relationship between accuracy and phonetic distance for adjectives (*z* = −1.47, *p* = 0.368, OR = 0.89, 95% CI[0.76, 1.04]). For Japonic-Sino-Tai languages, accuracy increased with shorter phonetic distances for nouns (*z* = −4.57,

**Table 2 Tukey-adjusted pairwise comparisons of accuracy in each language.**

| Contrast | Odds ratio | SE | 95% CI | z | p |
|---|---|---|---|---|---|
| Japanese/Mandarin | 1.01 | 0.07 | [0.82, 1.26] | 0.18 | 1.000 |
| Japanese/Thai | 0.92 | 0.06 | [0.74, 1.14] | −1.21 | 0.955 |
| Japanese/Polish | 0.91 | 0.06 | [0.74, 1.13] | −1.37 | 0.910 |
| Japanese/Russian | 0.97 | 0.07 | [0.79, 1.20] | −0.40 | 1.000 |
| Japanese/Ukrainian | 0.88 | 0.06 | [0.71, 1.09] | −1.83 | 0.663 |
| Japanese/French | 0.33 | 0.03 | [0.25, 0.42] | −13.91 | <0.0001 |
| Japanese/Romanian | 0.44 | 0.03 | [0.35, 0.55] | −11.35 | <0.0001 |
| Japanese/Spanish | 0.29 | 0.02 | [0.23, 0.38] | −15.19 | <0.0001 |
| Mandarin/Thai | 0.91 | 0.06 | [0.73, 1.13] | −1.39 | 0.902 |
| Mandarin/Polish | 0.90 | 0.06 | [0.73, 1.11] | −1.56 | 0.826 |
| Mandarin/Russian | 0.96 | 0.07 | [0.78, 1.19] | −0.58 | 1.000 |
| Mandarin/Ukrainian | 0.87 | 0.06 | [0.71, 1.08] | −2.02 | 0.528 |
| Mandarin/French | 0.32 | 0.03 | [0.25, 0.41] | −14.10 | <0.0001 |
| Mandarin/Romanian | 0.43 | 0.03 | [0.34, 0.54] | −11.56 | <0.0001 |
| Mandarin/Spanish | 0.29 | 0.02 | [0.23, 0.37] | −15.42 | <0.0001 |
| Thai/Polish | 0.99 | 0.07 | [0.80, 1.23] | −0.13 | 1.000 |
| Thai/Russian | 1.06 | 0.07 | [0.85, 1.31] | 0.83 | 0.996 |
| Thai/Ukrainian | 0.96 | 0.07 | [0.78, 1.19] | −0.59 | 1.000 |
| Thai/French | 0.35 | 0.03 | [0.28, 0.46] | −12.86 | <0.0001 |
| Thai/Romanian | 0.48 | 0.03 | [0.38, 0.60] | −10.15 | <0.0001 |
| Thai/Spanish | 0.32 | 0.03 | [0.25, 0.41] | −14.06 | <0.0001 |
| Polish/Russian | 1.07 | 0.08 | [0.86, 1.33] | 0.94 | 0.990 |
| Polish/Ukrainian | 0.97 | 0.07 | [0.78, 1.21] | −0.44 | 1.000 |
| Polish/French | 0.36 | 0.03 | [0.28, 0.46] | −12.74 | <0.0001 |
| Polish/Romanian | 0.48 | 0.04 | [0.38, 0.60] | −10.05 | <0.0001 |
| Polish/Spanish | 0.32 | 0.03 | [0.25, 0.41] | −13.96 | <0.0001 |
| Russian/Ukrainian | 0.91 | 0.06 | [0.73, 1.13] | −1.39 | 0.903 |
| Russian/French | 0.34 | 0.03 | [0.26, 0.43] | −13.52 | <0.0001 |
| Russian/Romanian | 0.45 | 0.03 | [0.36, 0.56] | −10.89 | <0.0001 |
| Russian/Spanish | 0.30 | 0.02 | [0.23, 0.39] | −14.78 | <0.0001 |
| Ukrainian/French | 0.37 | 0.03 | [0.29, 0.47] | −12.37 | <0.0001 |
| Ukrainian/Romanian | 0.50 | 0.04 | [0.40, 0.62] | −9.64 | <0.0001 |
| Ukrainian/Spanish | 0.33 | 0.03 | [0.26, 0.43] | −13.69 | <0.0001 |
| French/Romanian | 1.34 | 0.12 | [1.03, 1.75] | 3.44 | 0.017 |
| French/Spanish | 0.90 | 0.08 | [0.67, 1.20] | −1.14 | 0.968 |
| Romanian/Spanish | 0.67 | 0.06 | [0.51, 0.87] | −4.66 | <0.001 |

$p < 0.001$, OR = 0.77, 95% CI[0.68, 0.86]), but no evidence of a relationship between accuracy and phonetic distance was found for verbs ($z = -1.59$, $p = 0.301$, OR = 0.89, 95% CI[0.77, 1.03]) or adjectives ($z = -1.29$, $p = 0.480$, OR = 0.91, 95% CI[0.79, 1.05]). In contrast for Slavic languages, accuracy increased with shorter phonetic distances for adjectives ($z = -4.48$, $p < 0.001$, OR = 0.67, 95% CI[0.57, 0.80]), but no evidence of a relationship between accuracy and phonetic distance was found for nouns ($z = -2.10$, $p = 0.103$, OR = 0.86, 95% CI[0.76, 0.99]) or verbs ($z = 0.98$, $p = 0.699$, OR = 1.06, 95% CI[0.94, 1.20]). Tukey-adjusted pairwise comparisons revealed that for both nouns and verbs, the facilitative effect of shorter phonetic distance was greater for Romance languages compared to Slavic languages (Nouns: $z = 8.88$, $p < 0.001$, OR = 2.45, 95% CI[1.94, 3.11]; Verbs: $z = 9.49$, $p < 0.001$, OR = 2.64, 95% CI[2.08, 3.36]) and Japonic-Sino-Tai Languages (Nouns: $z = 8.59$, $p < 0.001$ OR = 2.18, 95% CI[1.76, 2.69]; Verbs: $z = 6.87$, $p < 0.001$, OR = 2.21, 95% CI[1.69, 2.90]). For adjectives, the effect of phonetic distance was greater for Slavic languages than Japonic-Sino-Tai Languages ($z = 2.71$, $p = 0.020$, OR = 1.35, 95% CI[1.04, 1.75]; see Fig. 5).

Critically, a significant intercept term indicated that overall, participants identified the meaning of the foreign word pairs significantly better than what would be expected by chance, even after controlling for phonetic distance from English ($\chi^2(1) = 114.31$, $p < 0.001$). Sidak-adjusted tests of simple effects confirmed that this was the case for all language groups, including Japonic-Sino-Tai ($z = 5.29$, $p < 0.001$, Estimate = 0.57, SE = 0.01, 95% CI[0.54, 0.60]), Slavic ($z = 5.87$, $p < 0.001$, Estimate = 0.58, SE = 0.01, 95% CI[0.55, 0.61]), and Romance languages ($z = 16.85$, $p < 0.001$, Estimate = 0.74, SE = 0.01, 95% CI[0.71, 0.76]). A main effect of Language Group ($\chi^2(2) = 278.42$, $p < 0.001$) showed that accuracy was significantly higher for Romance languages ($M = 73.6\%$) than Slavic languages ($M = 58.1\%$; $z = 14.91$, $p < 0.001$, OR = 0.50, 95% CI[0.45, 0.56]) and Japonic-Sino-Tai languages ($M = 57.4\%$; $z = 15.16$, $p < 0.001$, OR = 0.49, 95% CI[0.43, 0.54]).

**Individual differences in form-meaning mapping**. A measure of cross-linguistic form-meaning regularity was calculated by estimating the extent to which words with the same meaning had greater phonological overlap across words in different languages than words with opposite meanings (e.g., the relative distance between the Ukrainian word for "blunt" and the Japanese word for "blunt" vs. the Ukrainian word for "blunt" and the Japanese word for "sharp"). Consistent with the hypothesis that even arbitrary words retain some universal form-meaning regularities, we observed that phonetic distances among foreign language translations of words sharing a meaning were significantly shorter than distances among foreign words with opposite meanings ($\chi^2(1) = 10.94$, $p < 0.001$, Estimate = −0.01, SE = 0.002, 95% CI[−0.01, −0.003]). We found no evidence that the effect of meaning was moderated by language pair ($\chi^2(26) = 18.72$, $p = 0.85$) or part of speech ($\chi^2(2) = 3.75$, $p = 0.15$). The

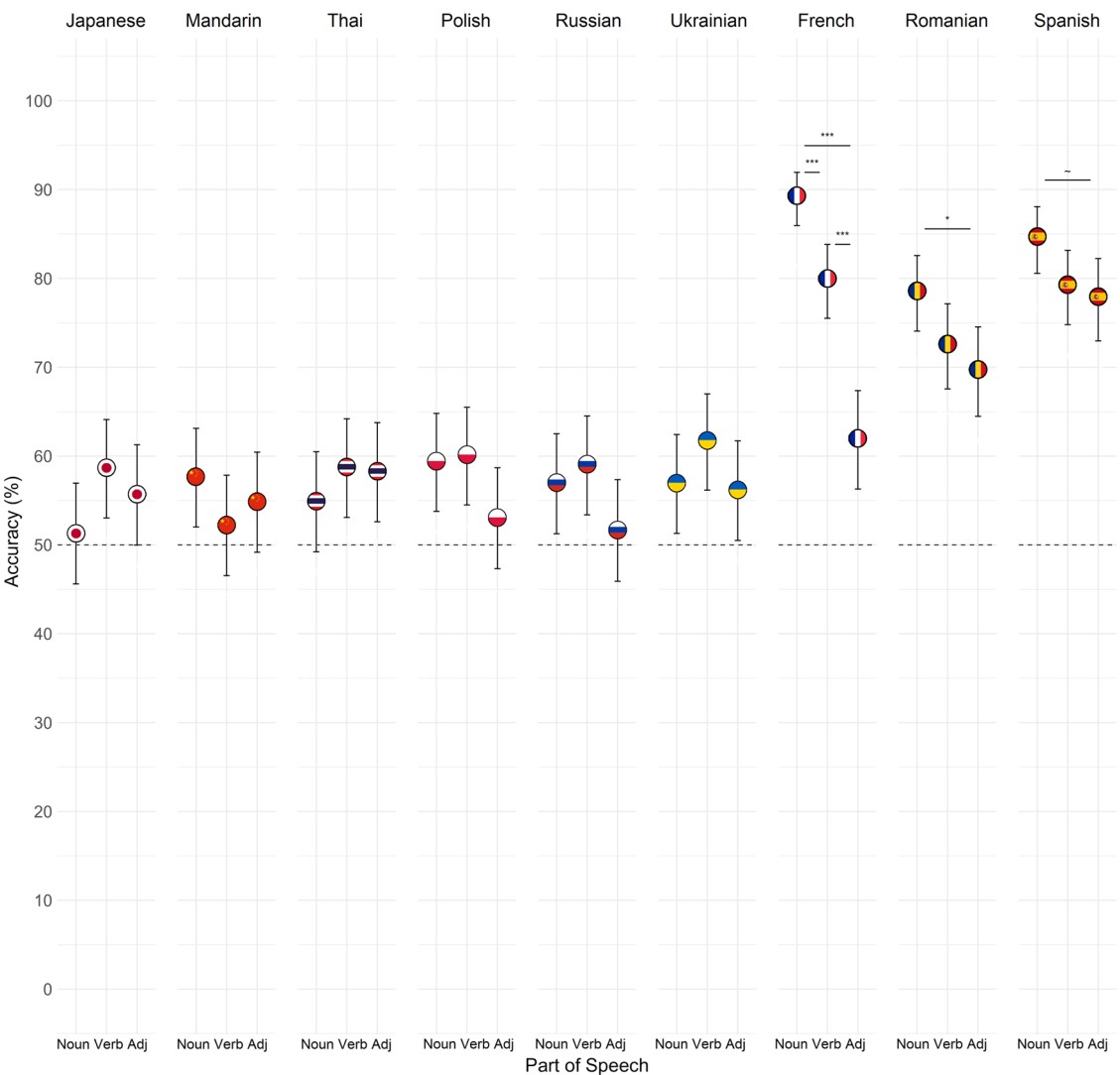

**Fig. 4 Mean accuracy by part of speech and language for native English speakers.** The meanings of noun-pairs were identified more accurately than adjective-pairs in the three Romance languages ($n = 44$ French, $n = 45$ Romanian, $n = 45$ Spanish). For French, meanings of nouns were identified more accurately than verbs, which were in turn identified more accurately than adjectives. We found no evidence that accuracy differed by part of speech for the other languages ($n = 45$ independent participants for Japanese, Mandarin, Polish, Ukrainian; $n = 44$ for Thai, Russian). Error bars represent 95% confidence intervals. $\sim p < 0.08$, $*p < 0.05$, $***p < 0.001$.

magnitude of difference in phonological overlap between same vs. opposite-meaning words was used to calculate form-meaning regularity scores for each word pair (averaged across comparisons with the six languages outside of the target pair's language group), with higher scores reflecting more consistent form-meaning correspondences.

Model tests indicated no evidence of heterogeneity of variances across language groups ($F(2) = 1.55$, $p = 0.213$) or parts of speech ($F(2) = 0.02$, $p = 0.981$), and no evidence of overdispersion ($\chi^2 = 1.003$, $p = 0.816$). A main effect of Verbal Working Memory ($\chi^2(1) = 24.80$, $p < 0.001$, OR $= 1.10$, 95% CI[1.06, 1.14]) indicates that, as predicted, accuracy was greater for individuals with better verbal working memory. There was a significant two-way interaction between Form-Meaning Regularity and Verbal Working Memory ($\chi^2(1) = 5.35$, $p < 0.021$, OR $= 1.04$, 95% CI[1.01, 1.08]), with Sidak-adjusted simple effects demonstrating that higher verbal working memory significantly increased accuracy for words with greater form-meaning regularity ($+1$ SD from the mean; $z = 5.23$, $p < 0.001$, OR $= 1.15$, 95% CI[1.09, 1.21]). We found no evidence that

verbal working memory increased accuracy for word pairs with less regularity ($-1$ SD from the mean: $z = 2.00$, $p = 0.088$, OR $= 1.06$, 95% CI[1.00, 1.11]; see Fig. 6). Higher form-meaning regularity was associated with greater accuracy for participants with better verbal working memory ($+1$ SD from mean: $z = 2.39$, $p = 0.034$, OR $= 1.07$, 95% CI[1.01, 1.13]), whereas we found no evidence that accuracy increased with form-meaning regularity for those with worse verbal memory ($-1$ SD from mean: $z = -0.545$, $p = 0.828$, OR $= 0.98$, 95% CI[0.93, 1.04]). A significant three-way interaction suggests that the association between Verbal Working Memory and Form-Meaning Regularity additionally varies by Part of Speech ($\chi^2(2) = 7.28$, $p = 0.026$). Simple effects reveal that for words with both greater and less form-meaning regularity, better verbal working memory was associated with greater accuracy for nouns (greater regularity: $z = 3.20$, $p = 0.003$, OR $= 1.17$, 95% CI[1.06, 1.29]; less regularity: $z = 2.88$, $p = 0.008$, OR $= 1.14$, 95% CI[1.04, 1.24]) and adjectives (greater regularity: $z = 3.05$, $p = 0.005$, OR $= 1.13$, 95% CI[1.04, 1.21]; less regularity: $z = 2.46$, $p = 0.027$, OR $= 1.12$, 95% CI[1.02, 1.22]). For verbs, the effect of verbal working

**Table 3 Tukey-adjusted pairwise comparisons of accuracy in each part of speech by language.**

| Language | Contrast | Odds ratio | SE | 95% CI | z | p |
|---|---|---|---|---|---|---|
| Japanese | Adjective/Noun | 1.19 | 0.19 | [0.82, 1.75] | 1.09 | 0.522 |
| | Adjective/Verb | 0.89 | 0.14 | [0.60, 1.30] | −0.75 | 0.736 |
| | Noun/Verb | 0.74 | 0.12 | [0.51, 1.09] | −1.84 | 0.157 |
| Mandarin | Adjective/Noun | 0.89 | 0.14 | [0.61, 1.31] | −0.70 | 0.763 |
| | Adjective/Verb | 1.11 | 0.18 | [0.76, 1.63] | 0.66 | 0.788 |
| | Noun/Verb | 1.25 | 0.20 | [0.85, 1.82] | 1.36 | 0.362 |
| Thai | Adjective/Noun | 1.15 | 0.19 | [0.78, 1.68] | 0.84 | 0.677 |
| | Adjective/Verb | 0.98 | 0.16 | [0.67, 1.44] | −0.12 | 0.992 |
| | Noun/Verb | 0.86 | 0.14 | [0.58, 1.25] | −0.96 | 0.602 |
| Polish | Adjective/Noun | 0.77 | 0.13 | [0.53, 1.13] | −1.59 | 0.251 |
| | Adjective/Verb | 0.75 | 0.12 | [0.51, 1.10] | −1.77 | 0.18 |
| | Noun/Verb | 0.97 | 0.16 | [0.66, 1.42] | −0.18 | 0.982 |
| Russian | Adjective/Noun | 0.81 | 0.13 | [0.55, 1.18] | −1.31 | 0.388 |
| | Adjective/Verb | 0.74 | 0.12 | [0.50, 1.09] | −1.83 | 0.158 |
| | Noun/Verb | 0.92 | 0.15 | [0.62, 1.35] | −0.52 | 0.86 |
| Ukrainian | Adjective/Noun | 0.97 | 0.16 | [0.66, 1.42] | −0.19 | 0.98 |
| | Adjective/Verb | 0.79 | 0.13 | [0.54, 1.17] | −1.41 | 0.337 |
| | Noun/Verb | 0.82 | 0.13 | [0.56, 1.20] | −1.22 | 0.443 |
| French | Adjective/Noun | 0.20 | 0.04 | [0.12, 0.31] | −8.27 | <0.0001 |
| | Adjective/Verb | 0.41 | 0.07 | [0.27, 0.62] | −5.07 | <0.0001 |
| | Noun/Verb | 2.09 | 0.43 | [1.29, 3.38] | 3.59 | 0.001 |
| Romanian | Adjective/Noun | 0.63 | 0.11 | [0.42, 0.95] | −2.66 | 0.023 |
| | Adjective/Verb | 0.87 | 0.15 | [0.58, 1.30] | −0.82 | 0.691 |
| | Noun/Verb | 1.39 | 0.24 | [0.92, 2.10] | 1.86 | 0.152 |
| Spanish | Adjective/Noun | 0.64 | 0.13 | [0.40, 1.02] | −2.25 | 0.072 |
| | Adjective/Verb | 0.92 | 0.17 | [0.60, 1.43] | −0.43 | 0.905 |
| | Noun/Verb | 1.45 | 0.28 | [0.92, 2.28] | 1.90 | 0.139 |

memory was significant for words with greater form-meaning regularity ($z = 3.15$, $p = 0.003$, OR = 1.15, 95% CI[1.05, 1.25]), whereas we found no evidence that accuracy increased with greater verbal working memory for words with less regularity ($z = -1.87$, $p = 0.134$, OR = 0.92, 95% CI[0.85, 1.00]).

A significant intercept term confirmed that participants identified the meaning of the foreign word pairs significantly better than chance even when controlling for verbal working memory and form-meaning regularity ($\chi^2(1) = 129.89$, $p < 0.001$), and that this was the case for all language groups, including Japonic-Sino-Tai ($z = 4.24$, $p < 0.001$, Estimate = 0.56, SE = 0.01, 95% CI[0.54, 0.59]), Slavic ($z = 5.77$, $p < 0.001$, Estimate = 0.58, SE = 0.01, 95% CI[0.56, 0.61]), and Romance languages ($z = 20.22$, $p < 0.001$, Estimate = 0.78, SE = 0.01, 95% CI[0.76, 0.80]). There was additionally a significant main effect of Language Group ($\chi^2(2) = 554.70$, $p < 0.001$), with higher accuracy for Romance Indo-European languages ($M = 77.7\%$, 95% CI[75.5, 79.7]) than Slavic Indo-European languages ($M = 58.3\%$, 95% CI[55.5, 61.1]; $z = -19.83$, $p < 0.001$, OR = 0.40, 95% CI[0.36, 0.45]) and Japonic-Sino-Tai languages ($M = 56.1\%$, 95% CI[53.3, 58.8]; $z = -22.16$, $p < 0.001$, OR = 0.37, 95% CI[0.33, 0.41]). Lastly, there was a significant three-way interaction between Form-Meaning Regularity, Part of Speech, and Language Group ($\chi^2(4) = 60.30$, $p < 0.001$). Tukey-adjusted pairwise comparisons revealed that accuracy was significantly higher for nouns compared to adjectives for Romance ($z = -2.42$, $p = 0.041$, OR = 0.65, 95% CI[0.44, 0.99]) and Slavic ($z = -2.71$, $p = 0.018$, OR = 0.64, 95% CI[0.44, 0.94]) words with greater form-meaning regularity. Accuracy was additionally higher for verbs compared to adjectives for Romance words with greater regularity ($z = -4.27$, $p < 0.001$, OR = 0.49, 95% CI[0.33, 0.72]). Accuracy was significantly higher for nouns compared to verbs ($z = 6.08$, $p < 0.001$, OR = 3.14, 95% CI[2.02, 4.88]) and adjectives ($z = -6.19$, $p < 0.001$, OR = 0.31, 95% CI[0.20, 0.49]) for Romance words with less regularity. We found no evidence that

accuracy differed by part of speech for the remaining languages and degrees of form-meaning regularity ($ps > 0.05$; see Table S2).

**Replication with native Spanish speakers: effects of language and part of speech.** To replicate these findings and determine the generalizability of effects to a different population, the experiment was replicated with 46 native Spanish speakers with minimal foreign language experience, including with English. In addition to completing the antonym word pair task in English (a Germanic language), participants were tested in Japanese and Polish (the Japonic-Sino-Tai and Slavic languages with the median effect size among the three languages in their respective groups). Due to substantial overlap among Romance languages, Spanish speakers were not tested in French or Romanian.

Model tests indicated no evidence of heterogeneity of variances across languages ($F(2) = 1.18$, $p = 0.308$) or parts of speech ($F(2) = 2.01$, $p = 0.135$), and no evidence of overdispersion ($\chi^2 = 1.005$, $p = 0.704$). As in the first study, participants were able to correctly identify the meanings of foreign language antonym pairs better than what would be expected by chance ($\chi^2(1) = 36.52$, $p < 0.001$), and this was the case in all three languages (Japanese: $z = 2.16$, $p = 0.031$, Estimate = 0.55, SE = 0.02, 95% CI[0.50, 0.59]; Polish: $z = 4.35$, $p < 0.001$, Estimate = 0.59, SE = 0.02, 95% CI[0.55, 0.63]; English: $z = 8.78$, $p < 0.001$, Estimate = 0.70, SE = 0.02, 95% CI[0.66, 0.74]). A main effect of Language ($\chi^2(2) = 68.95$, $p < 0.001$) revealed that accuracy was significantly higher for English compared to both Polish ($z = -5.98$, $p < 0.001$, OR = 0.62, 95% CI[0.52, 0.75]) and Japanese ($z = -8.28$, $p < 0.001$, OR = 0.52, 95% CI[0.43, 0.62]), and for Polish compared to Japanese ($z = -2.83$, $p = 0.013$, OR = 0.83, 95% CI[0.72, 0.97]). Finally, there was a significant interaction between Language and Part of Speech ($\chi^2(4) = 44.24$, $p < 0.001$). Similar to the pattern observed in the first study, accuracy in English was significantly higher for nouns ($M = 0.80$,

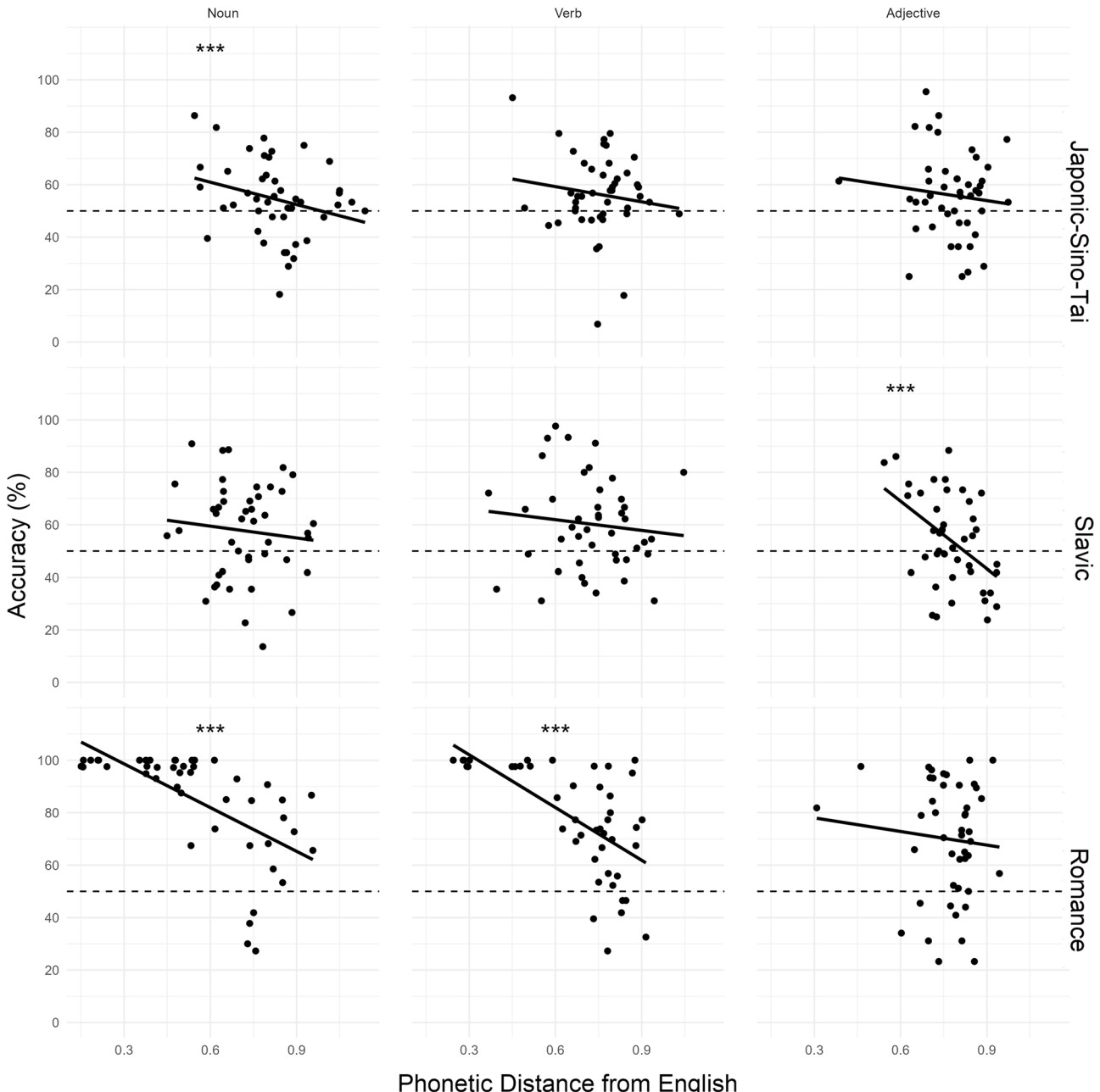

**Fig. 5 Mean accuracy by phonetic distance from English, part of speech (nouns, verbs, adjectives), and language group (Japonic-Sino-Tai, Slavic, Romance) for native English speakers.** The meanings of foreign word pairs were identified more accurately with shorter phonetic distances to their English translations. For nouns and verbs, the facilitative effect of shorter phonetic distance was greater for Romance languages than Slavic and Japonic-Sino-Tai languages. For adjectives, the facilitative effect of shorter phonetic distance was greater for Slavic than Japonic-Sino-Tai languages. Dots represent mean accuracy for individual items ($n = 134$ independent participants).

SE = 0.03) and verbs ($M = 0.70$, SE = 0.03) compared to adjectives ($M = 0.57$, SE = 0.03; Nouns vs. Adjectives: $z = -4.84$, $p < 0.001$, OR = 0.33, 95% CI[0.19, 0.56]; Verbs vs. Adjectives: $z = -2.65$, $p = 0.022$, OR = 0.56, 95% CI[0.33, 0.93]). Accuracy for English nouns was marginally higher than for verbs ($z = 2.29$, $p = 0.058$, OR = 1.71, 95% CI[0.99, 2.96]). We found no evidence that accuracy varied by part of speech for Japanese (Nouns vs. Adjectives: $z = -0.03$, $p = 0.999$, OR = 0.99, 95% CI[0.62, 1.59]; Verbs vs. Adjectives: $z = -1.22$, $p = 0.44$, OR = 0.78, 95% CI[0.49, 1.25]; Nouns vs. Verbs: $z = -1.19$, $p = 0.457$, OR = 0.79, 95% CI[0.49, 1.26]) or Polish (Nouns vs. Adjectives: $z = 0.02$, $p > 0.999$, OR = 1.00, 95% CI[0.63, 1.60]; Verbs vs. Adjectives: $z = 0.42$, $p = 0.907$, OR = 1.09, 95% CI[0.68, 1.74],

Nouns vs. Verbs: $z = 0.40$, $p = 0.914$, OR = 1.08, 95% CI[0.68, 1.73]).

**Effects of language and phonetic distance from Spanish.** A significant interaction between Language and Part of Speech ($\chi^2(4) = 27.80$, $p < 0.001$) revealed that for English, average distance from Spanish was significantly shorter for nouns ($M = 0.55$, SE = 0.03) and verbs ($M = 0.65$, SE = 0.03) than for adjectives ($M = 0.79$, SE = 0.03; Nouns vs. Adjectives: $t(126) = 5.06$, $p < 0.001$, Estimate = 0.24, SE = 0.05, 95% CI[0.13, 0.36]; Verbs vs. Adjectives: $t(126) = 2.92$, $p = 0.012$, Estimate = 0.14, SE = 0.05, 95% CI[0.03, 0.26]). We found no evidence that phonetic distance from Spanish differed between nouns and verbs in

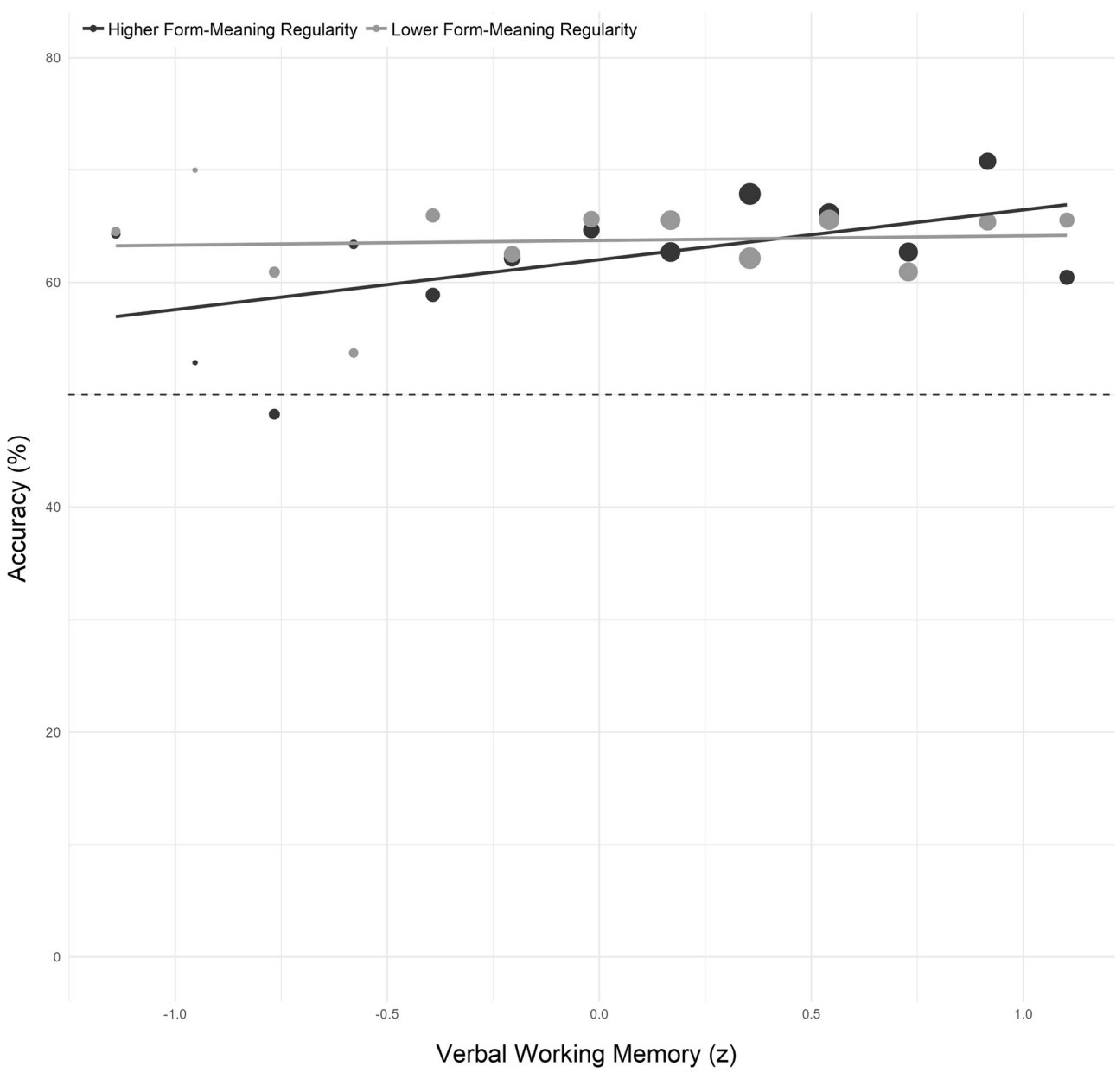

**Fig. 6 Mean accuracy by mean centered verbal working memory and form-meaning regularity (higher: +1 SD, lower: −1 SD) for native English speakers.** The meanings of foreign word pairs were identified more accurately among participants with higher verbal working memory. The facilitative effect of better verbal working memory was greater for word pairs with high (black) than low (gray) form-meaning regularity. Dot sizes represent the number of independent participants contributing to each data point (min = 2, max = 19; total = 122).

English ($t(126) = -2.14$, $p = 0.085$, Estimate $= -0.10$, SE $= 0.05$, 95% CI[−0.22, 0.01]) or by any part of speech in Japanese (Nouns vs. Adjectives: $t(126) = -0.11$, $p = 0.993$, Estimate $= -0.01$, SE $= 0.05$, 95% CI[−0.12, 0.11]; Verbs vs. Adjectives: $t(126) = 0.95$, $p = 0.611$, Estimate $= 0.05$, SE $= 0.05$, 95% CI[−0.07, 0.16]; Nouns vs. Verbs: $t(126) = 1.06$, $p = 0.543$, Estimate $= 0.05$, SE $= 0.05$, 95% CI[−0.06, 0.17]) or Polish (Nouns vs. Adjectives: $t(126) = -1.73$, $p = 0.197$, Estimate $= -0.08$, SE $= 0.05$, 95% CI[−0.20, 0.03]; Verbs vs. Adjectives: $t(126) = -1.35$, $p = 0.369$, Estimate $= -0.07$, SE $= 0.05$, 95% CI[−0.18, 0.05]; Nouns vs. Verbs: $t(126) = 0.38$, $p = 0.923$, Estimate $= 0.02$, SE $= 0.05$, 95% CI[−0.1, 0.13]).

Model tests indicated no evidence of heterogeneity of variances across languages ($F(2) = 1.11$, $p = 0.328$) or parts of speech ($F(2) = 2.53$, $p = 0.080$), and no evidence of overdispersion ($\chi^2 = 1.003$, $p = 0.792$). Similar to the first study, accuracy increased with shorter phonetic distances from Spanish ($\chi^2(1) = 50.47$, $p < 0.001$, Estimate $= -0.39$, SE $= 0.06$, 95% CI[−0.50, −0.28]), which was qualified by a significant three-way interaction between Language, Part of Speech, and Phonetic Distance ($\chi^2(4) = 20.04$, $p < 0.001$). For English, Sidak-adjusted tests of simple effects revealed that accuracy significantly increased with shorter phonetic distances for nouns ($z = -5.45$, $p < 0.001$, OR $= 0.58$, 95% CI[0.48, 0.71]) and verbs ($z = -5.37$, $p < 0.001$, OR $= 0.52$, 95% CI[0.41, 0.66]). We found no evidence that phonetic distance affected accuracy for adjectives ($z = 0.29$, $p = 0.988$, OR $= 1.07$, 95% CI[0.68, 1.68]). For Japanese, accuracy increased with shorter phonetic distances for nouns ($z = -2.56$,

$p = 0.031$, OR $= 0.58$, 95% CI[0.39, 0.88]), but we found no evidence that phonetic distance affected accuracy for verbs ($z = -0.01$, $p > 0.999$, OR $= 1.00$, 95% CI[0.71, 1.40] or adjectives ($z = -0.82$, $p = 0.797$, OR $= 0.85$, 95% CI[0.59, 1.24]). In contrast for Polish, accuracy increased with shorter phonetic distances for adjectives ($z = -6.38$, $p < 0.001$, OR $= 0.43$, 95% CI[0.33, 0.56]) and verbs ($z = -4.33$, $p < 0.001$, OR $= 0.57$, 95% CI[0.44, 0.73]), but we found no evidence that phonetic distance affected accuracy for nouns ($z = -1.78$, $p = 0.208$, OR $= 0.76$, 95% CI[0.57, 1.03]).

Critically, a significant intercept term indicated that overall, participants identified the meaning of the foreign word pairs significantly better than would be expected by chance, even after controlling for phonetic distance from Spanish ($\chi^2(1) = 37.13$, $p < 0.001$). Sidak-adjusted tests of simple effects confirmed that this was the case for all languages, including English ($z = 6.52$, $p < 0.001$, Estimate $= 0.67$, SE $= 0.02$, 95% CI[0.62, 0.71]), Japanese ($z = 2.95$, $p = 0.009$, Estimate $= 0.56$, SE $= 0.02$, 95% CI[0.52, 0.59]), and Polish ($z = 4.17$, $p < 0.001$, Estimate $= 0.58$, SE $= 0.02$, 95% CI[0.54, 0.62]). A main effect of Language ($\chi^2(2) = 20.43$, $p < 0.001$) showed that accuracy was significantly higher for English ($M = 66.8\%$) than Polish ($M = 58.0\%$; $z = -3.67$, $p < 0.001$, OR $= 0.69$, 95% CI[0.54, 0.87]) and Japanese ($M = 55.7\%$; $z = -4.56$, $p < 0.001$, OR $= 0.63$, 95% CI[0.49, 0.80]). We found no evidence that accuracy differed between Polish and Japanese after controlling for phonetic distance from Spanish ($z = -1.29$, $p = 0.402$, OR $= 0.91$, 95% CI[0.77, 1.08]).

**Individual differences in form-meaning mapping**. Our measure of Form-Meaning Regularity was recalculated to exclude phonetic distances from the native language of Spanish and to include phonetic distances to the foreign language of English. As in the first study, we observed that phonetic distances among foreign language translations of words sharing a meaning were significantly shorter than distances among foreign words with opposite meanings ($\chi^2(1) = 5.51$, $p = 0.019$, Estimate $= -0.008$, SE $= 0.003$, 95% CI[$-0.01$, $-0.001$]). We found no evidence that the effect of meaning was moderated by language pair ($\chi^2(8) = 4.09$, $p = 0.848$) or part of speech ($\chi^2(2) = 1.06$, $p = 0.588$).

Model tests indicated no evidence of heterogeneity of variances across languages ($F(2) = 0.24$, $p = 0.785$) or parts of speech ($F(2) = 1.44$, $p = 0.237$), and no evidence of overdispersion ($\chi^2 = 1.02$, $p = 0.272$). As predicted, a main effect of Form-Meaning Regularity indicates that accuracy was higher for words with more consistent form-meaning mappings across unrelated languages ($\chi^2(1) = 70.73$, $p < 0.001$, OR $= 1.41$, 95% CI[1.29, 1.52]; see Fig. 7). In contrast to the first study, however, we did not find evidence of a main effect ($\chi^2(1) = 0.07$, $p = 0.795$, OR $= 1.01$, 95% CI[0.93, 1.09]) or interaction with Verbal Working Memory ($\chi^2(2) = 0.33$, $p = 0.850$, OR $= 1.02$, 95% CI[0.96, 1.09]). Although the pattern of effects resembled that of the first study, with a numerically greater positive effect of higher verbal working memory for words with greater form-meaning regularity ($z = 0.65$, $p = 0.762$, OR $= 1.03$, 95% CI[0.89, 1.09]) than those with less form-meaning regularity ($z = -0.26$, $p = 0.957$, OR $= 0.98$, 95% CI[0.93, 1.15]), neither trend approached significance.

A significant intercept term confirmed that participants identified the meaning of the foreign word pairs significantly better than chance even when controlling for verbal working memory and form-meaning regularity ($\chi^2(1) = 35.19$, $p < 0.001$), and that this was the case for all languages, including English ($z = 7.37$, $p < 0.001$, Estimate $= 0.69$, SE $= 0.02$, 95% CI[0.64,

0.73]), Japanese ($z = 2.5$, $p = 0.037$, Estimate $= 0.56$, SE $= 0.02$, 95% CI[0.51, 0.6]), and Polish ($z = 5.09$, $p < 0.001$, Estimate $= 0.61$, SE $= 0.02$, 95% CI[0.57, 0.65]). There was additionally a significant main effect of Language ($\chi^2(2) = 40.18$, $p < 0.001$), with higher accuracy for English ($M = 68.8\%$, 95% CI[64.1, 73.1]) than both Polish ($M = 61.3\%$, 95% CI[57.0, 65.4]; $z = -3.62$, $p < 0.001$, OR $= 0.72$, 95% CI[0.58, 0.89]) and Japanese ($M = 55.6\%$, 95% CI[51.2, 59.8]; $z = -6.25$, $p < 0.001$, OR $= 0.57$, 95% CI[0.46, 0.70]), and higher accuracy for Polish than Japanese ($z = -3.37$, $p = 0.002$, OR $= 0.79$, 95% CI[0.67, 0.93]). Lastly, as in the first study, there was a significant three-way interaction between Form-Meaning Regularity, Part of Speech, and Language ($\chi^2(4) = 26.48$, $p < 0.001$). Tukey-adjusted pairwise comparisons revealed that accuracy was significantly higher for nouns compared to adjectives for English words with both greater ($z = -3.42$, $p = 0.002$, OR $= 0.36$, 95% CI[0.18, 0.72]) and less form-meaning regularity ($z = -2.82$, $p = 0.013$, OR $= 0.38$, 95% CI[0.17, 0.85]). Accuracy was additionally higher for Polish adjectives compared to verbs with greater form-meaning regularity ($z = 3.41$, $p = 0.002$, OR $= 2.69$, 95% CI[1.36, 5.33]) and for Japanese verbs relative to nouns with less form-meaning regularity ($z = -2.38$, $p = 0.045$, OR $= 0.54$, 95% CI[0.29, 0.99]; see Table S3).

## Discussion
Over 7000 languages are spoken around the world today, each with distinct sounds, words, and rules shaped by unique sociological and geopolitical trajectories. Yet all human languages evolved with a common set of communicative tools to represent the same external world, giving rise to associations between form and meaning that are remarkably consistent across otherwise unrelated linguistic systems. Studying form-meaning mapping can therefore inform not only how language evolved over generations, but also how language is represented and decoded by the individual (e.g., the mechanisms underlying the acquisition and use of linguistic symbols). The goal of the present study was to examine the relationship between form and meaning in natural language and to identify the cognitive and linguistic predictors of sensitivity to form-meaning regularities. Contrary to the view of language as a system built on arbitrary form-meaning connections, we observed that words that shared a meaning across languages had greater phonological overlap than words that did not share a meaning. Critically, our measure of form-meaning regularity reflected consistency in covariation between phonology and semantics across distinct language branches and even language families (e.g., Japanese and Russian), providing compelling support for relatively "universal" links between the sounds and meanings of words[9,42,62]. Behavioral findings revealed that native English and Spanish speakers were able to correctly identify the meanings of foreign language antonym pairs significantly better than would be expected by chance. The ability to correctly infer meaning from real foreign language words demonstrates that there are systematic form-meaning regularities that transcend linguistic boundaries.

The present study also demonstrates that the ability to capitalize on form-meaning regularities is tied to cognitive function and further speaks to the link between language and cognition. The association between form-meaning mapping and verbal working memory observed with native English speakers suggests that, rather than being a mere artifact or niche curiosity, sensitivity to relations between form and meaning (and possibly iconicity) in language may be indicative of more general cognitive function. Verbal working memory may play a role in word learning and recognition by enabling listeners to bind

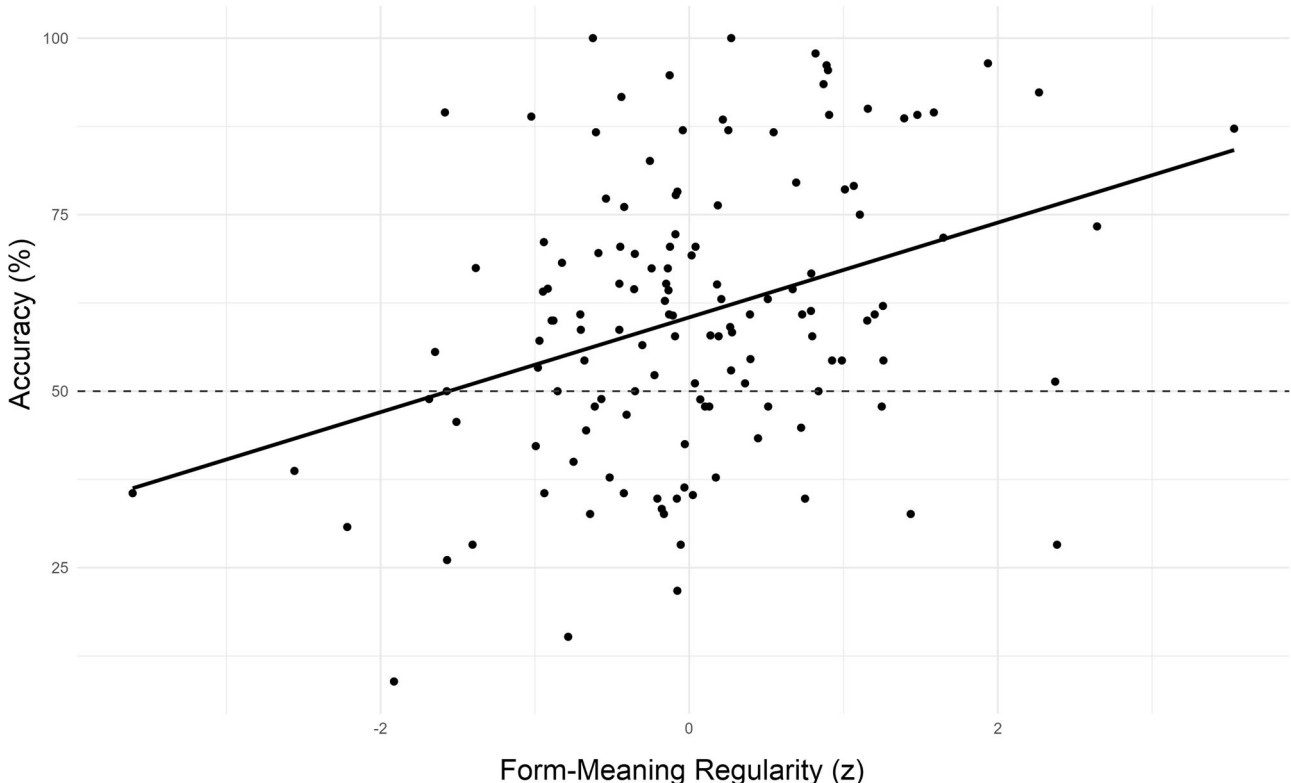

**Fig. 7 Mean accuracy by mean centered form-meaning regularity for native Spanish speakers.** The meanings of foreign word pairs were identified more accurately for words with more consistent form-meaning mappings across languages. Dots represent individual items ($n = 43$ independent participants).

information about a word's phonological form with perceptual and semantic characteristics of its referent[49,50,63]. Better verbal working memory was not only predictive of overall accuracy, it also specifically facilitated semantic decoding of words that had more consistent form-meaning regularities across foreign languages. This finding suggests that better verbal working memory may enhance sensitivity to covarying relationships between form and meaning in known languages, which can then be generalized to unknown languages.

Findings from the replication study, however, indicate that the effect of verbal working memory may differ depending on the language population. Specifically, we observed that while both native English speakers and especially native Spanish speakers were sensitive to the degree of form-meaning regularity across languages, the effect of verbal working memory was limited to native English speakers. It is possible that reliance on verbal working memory resources varies depending on prior language experience. The linguistic characteristics of English and Spanish differ in several ways that may affect how much its speakers rely on verbal working memory during auditory comprehension. For instance, the relationship between the spelling and pronunciation of words (i.e., orthographic transparency) is more consistent in Spanish relative to English, which may tax working memory differently. Likewise, because English is characterized by relatively more complex syllable structures (e.g., many syllables that begin with consonant clusters such as "splash" and "tree") and vowel systems (i.e., a larger number of vowel sounds) relative to Spanish, English speakers may rely on verbal working memory to a greater extent when listening to spoken words. Moreover, while both populations included participants with minimal foreign language experience, speakers of non-English languages are likely to have greater exposure to foreign languages in general (and English in particular). To the extent that more frequent multilingual exposure increases

sensitivity to linguistic regularities, the ability to pick up on associations between form and meaning may become less contingent on verbal working memory resources. Finally, this discrepancy between groups may be attributable to the relatively smaller sample size and power of the replication study ($N = 46$ native Spanish speakers compared to $N = 134$ native English speakers), as well as lower verbal working memory (digit span) scores for the native Spanish speakers ($M = 13.21$, $SD = 2.69$) compared to the native English speakers ($M = 16.50$, $SD = 2.92$, $t(79.58) = 6.75$, $p < 0.001$). Future work incorporating more detailed language experience measures, larger samples, and more diverse language populations will be critical to elucidate the relationship between verbal working memory and sensitivity to form-meaning regularities.

While the effect of verbal working memory varied across native language populations (native English vs. Spanish speakers), we found no statistically significant evidence that the relationship between verbal working memory and form-meaning mapping varied across foreign languages among native English speakers. This demonstrates that the benefits of better working memory (when observed) are not limited to languages that are more closely related to the native tongue (e.g., French-English vs. Japanese-English). Phonetic distance from the native language, did, however, exert an independent effect on accuracy for both English and Spanish speakers. The ability to correctly identify the meanings of foreign word pairs increased with greater phonological overlap between the foreign words and their native translations. The significantly higher accuracy for Romance Indo-European languages (among English speakers) and English, a Germanic Indo-European language (among Spanish speakers) relative to Slavic Indo-European and Japonic-Sino-Tai languages is therefore likely to stem from relatively greater phonological overlap between Romance and Germanic languages.

There are two notable insights that can be gained from the effects of phonetic distance from the native language. First, we found that participants identified the meanings of foreign words significantly better than chance even among more linguistically distant languages (i.e., Japonic-Sino-Tai and Slavic Indo-European languages) and even after statistically controlling for phonetic distance from the native language. These findings suggest that the ability to infer the meanings of foreign words cannot be fully attributed to direct cross-linguistic transfer at the lexical level (e.g., across cognates that share both form and meaning or near-cognates that share a meaning and have similar forms), offering further support for the exploitation of sublexical form-meaning regularities across languages. Second, we observed that the degree of phonological overlap between English and Romance languages varied by part of speech, and that these patterns coincided with effects of part of speech on accuracy (similar to effects observed by D'Anselmo et al.[12]). Specifically, both phonological overlap with English and accuracy were higher for nouns than verbs and for verbs than adjectives. To the extent that form-meaning regularities in language originate from evoking perceptual features of referents through speech[4–6], the observed differences in cross-linguistic overlap may be partly due to the fact that nouns are often more concrete[64] and thus easier to evoke relative to verbs and adjectives. It is likely no coincidence that this also reflects the general order of acquisition during children's language development (i.e., concrete nouns, followed by verbs and adjectives). Just as we observed greater accuracy for words that mapped form to meaning more consistently across languages, prior work has shown that both children[65,66] and adults[67] benefit from linguistic iconicity during the acquisition of novel words. Together, these findings suggest that the capacity to form perceptual associations between words and their referents may play a key role in shaping not only the words that exist in a language, but also how those words are subsequently learned.

**Limitations**. The present findings are consistent with earlier work on sound symbolism suggesting that iconicity in language may be more pervasive than is typically assumed[2,3,7–14,24]. It is important to note, however, that cross-linguistic form-meaning regularities may emerge from causes other than iconic relations between the sounds and meanings of words. In addition to historical relations between languages that can explain many cross-linguistic similarities, there may be other linguistic or communicative constraints that could give rise to non-arbitrary relations between form and meaning. So long as such regularities are present, listeners could rely on learned associations between particular sounds and meanings in their own languages to infer the meanings of unfamiliar words, even in the absence of linguistic iconicity. For instance, a native English speaker may make the connection between the /b/ sound in the English word "blunt" and its Japanese translation "*nibui*" without any resemblance between either word and the shared referent. Future research directly assessing the perceived similarity between particular sounds (e.g., /b/) and qualities of their referents (e.g., the sensory experience of bluntness) would provide stronger evidence that the observed effects stem from sound symbolism and iconicity in language.

## Conclusion
The present study provides evidence of regularities in sound-meaning mappings across nine natural languages. We demonstrate that the ability to extract meaning from form may be predicted by how consistently sounds covary with meanings across languages, as well as by individual differences in verbal working memory. We conclude that language is a representation of the external world that can become infused with our lived experiences.

## Data availability
The data that support the findings of this study can be accessed at https://osf.io/4ez8v/.

## Code availability
The code used to analyze the reported data can be accessed at https://osf.io/4ez8v/. The data were analyzed using the LingPy Python library, as well as *lme4*, *lmerTest*, and *emmeans* packages in R (version 4.1.2).

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

## Acknowledgements

This research was supported in part by the Eunice Kennedy Shriver National Institute of Child Health & Human Development of the National Institutes of Health under Award Number R01HD059858 to V.M. The content is solely the responsibility of the authors and does not necessarily represent the official views of the National Institutes of Health. The funders had no role in study design, data collection and analysis, decision to publish, or preparation of the manuscript. The authors thank Ashley Chung-Fat-Yim, Matias Fernandez-Duque, Angelika Labno, Siqi Ning, Daniel Ozernyi, and Sirada Rochanavibhata for their assistance with constructing the foreign language stimuli.

## Author contributions

S.H. designed the research, collected and analyzed the data, and wrote the paper. V.M. designed the research, wrote the paper, and supervised the project.

## Competing interests

The authors declare no competing interests.
