## [Peer Review File · Communications Psychology]

19th Jan 23

Dear Dr Hayakawa,

Thank you for your patience during the peer-review process. Your manuscript titled "How the mind finds meaning in the sounds of language" has now been seen by 3 reviewers, whose comments are appended below. You will see that they find your work of some potential interest. However, they have raised quite substantial concerns that must be addressed. In light of these comments, we cannot accept the manuscript for publication, but would be interested in considering a revised version that fully addresses these serious concerns.

We hope you will find the Reviewers' comments useful as you decide how to proceed. Should additional work allow you to address these criticisms, we would be happy to look at a substantially revised manuscript. If you choose to take up this option, please highlight all changes in the manuscript text file, and provide a detailed point-by-point reply to the reviewers.

Editorially, we consider it particularly important:

- (1) To provide additional data that tests the same effect in different language groups as suggested by Reviewer #1.
- (2) To conduct additional analyses as suggested by Reviewer #3 and to address the main concern voiced by Reviewer #1, that is, the degree to which the results allow inferences on the role of sound symbolism.
- (3) To provide a sufficiently caveated interpretation of the findings and a discussion of alternative explanations, as suggested by the reviewers.
- (4) To expand on the presentation of the methods. As you already plan to share data and code, we would strongly encourage you to also make the stimulus material available in the same space. If you decide against this, please provide a more comprehensive description in the Methods and a list of stimuli in the Supplemental Materials.
- (5) To incorporate the reviewer's literature suggestions to the extent that they help to better frame and interpret your study.

If the revision process takes significantly longer than five months, we will be happy to reconsider your paper at a later date, provided it still presents a significant contribution to the literature at that stage.

We understand that due to the current global situation, the time required for revision may be longer than usual. We would appreciate it if you could keep us informed about an estimated timescale for resubmission, to facilitate our planning. Of course, if you are unable to estimate, we are happy to accommodate necessary extensions nevertheless.

Please use the following link to submit your revised manuscript, point-by-point response to the Reviewers' comments with a list of your changes to the manuscript text (which should be in a separate document to any cover letter) and any completed checklist:

[link redacted]

Please do not hesitate to contact me if you have any questions or would like to discuss the required revisions further. Thank you for the opportunity to review your work.

Best regards,

Jennifer Bellingtier

Jennifer Bellingtier, PhD
Senior Editor
Communications Psychology

EDITORIAL POLICIES AND FORMATTING

Editorial Policy: [Policy requirements](https://www.nature.com/documents/nr-editorial-policy-checklist.pdf) (Download the link to your computer as a PDF.)

Furthermore, please align your manuscript with our format requirements, which are summarized on the following checklist:

[Communications Psychology formatting checklist](https://www.nature.com/documents/commsj-psychol-style-formatting-checklist-article.pdf)

and also in our style and formatting guide [Communications Psychology formatting guide](https://www.nature.com/documents/commsj-psychol-style-formatting-guide-accept.pdf) .

* **CODE AVAILABILITY:** All Communications Psychology manuscripts must include a section titled "Code Availability" at the end of the methods section. In the event of publication, we require that the custom analysis code supporting your conclusions is made available in a publicly accessible repository; please choose a repository that provides a DOI for the code; the link to the repository and the DOI must be included in the Code Availability statement. Publication as Supplementary Information will not suffice. We ask you to prepare and upload code at this stage, to avoid delays later on in the process.

* **DATA AVAILABILITY:**

All Communications Psychology research manuscripts must include a section titled "Data Availability" at the end of the Methods section or main text (if no Methods). More information on this policy, is available at <http://www.nature.com/authors/policies/data/data-availability-statements-data-citations.pdf>.

At a minimum the Data availability statement must explain how the data can be obtained and whether there are any restrictions on data sharing. Communications Psychology strongly endorses open sharing of data. If you do make your data openly available, please include in the statement:

We recommend submitting the data to discipline-specific, community-recognized repositories, where possible and a list of recommended repositories is provided at <http://www.nature.com/sdata/policies/repositories>.

If a community resource is unavailable, data can be submitted to generalist repositories such as [figshare](https://figshare.com/) or [Dryad Digital Repository](http://datadryad.org/). Please provide a unique identifier for the data (for example a DOI or a permanent URL) in the data availability statement, if possible. If the repository does not provide identifiers, we encourage authors to supply the search terms that will return the data. For data that have been obtained from publicly available sources, please provide a URL and the specific data product name in the data availability statement. Data with a DOI should be further cited in the methods reference section.

REVIEWER EXPERTISE:

Reviewer #1 Psycholinguistics

Reviewer #2 Psycholinguistics

Reviewer #3 Psycholinguistics

Reviewer #1 (Remarks to the Author):

This study addresses an important question in language science, namely the extent to which meaning can be gleaned from word forms alone, even in languages unknown to the learner (“if, and when, listeners can infer the meanings of foreign words based on their phonological forms alone”, p. 2). I find this motivation for the study sufficiently strong to justify the implemented methodological choices and the interpretation of the results. However, the study also tries to make a connection with sound symbolism which (regardless of the evidence we have for that phenomenon) is somewhat weaker and disconnected with the first motivation. Evidence for the ability to correctly guess meaning based on form in a foreign language may be evidence that there are systematic form-meaning regularities across languages, but that is not necessarily evidence for sound symbolism or iconicity: there could be a host of other reasons, ranging from historical relations between languages (which the authors do discuss) to other constraints that make the form-meaning mappings non-arbitrary (or less arbitrary than chance) and that would explain the observed results.

More specifically, I find the equation of cross-linguistic form-meaning regularity and sound symbolism problematic, in the absence of a discussion or argument that sound symbolism is the only or most likely account of cross-linguistic form-meaning regularity. This cannot just be assumed but has to be motivated, or least it has to be explicitly recognized that it is an assumption that may be overturned by new theoretical ideas on what could constrain form-meanings mappings to be non-arbitrary, beyond resemblance between sound and meaning.

Concretely, the authors write: “A measure of cross-linguistic form-meaning regularity (henceforth referred to as “sound symbolism”) was calculated by estimating the extent to which words with the same meaning had greater phonological overlap across words in different languages than words with opposite meanings.” But this requires showing independently (in a triangulation procedure) that there is sound-symbolism at least in one of the languages at issue: i.e., that ‘sharp’ resembles (sound-symbolically) to the meaning [sharp], or that ‘surudoï’ does, or both. The current measure is not specific enough and will capture all sorts of other non-arbitrary relations between forms across languages for words that share the same meanings: indeed, the strongest effects are found for historically related languages in this study, i.e., languages that share a common origin of large portions of the vocabulary through the common Latin root. I agree with the authors that their results indicate that “the ability to infer the meanings of foreign words cannot be fully attributed to direct cross-linguistic transfer at the lexical level”, but that does not automatically validate an explanation based on sound symbolism.

It would be important to know more about how the 45 English antonym word pairs were chosen in the first place, and for how many there actually was/wasn’t sound symbolism of the ‘sharp’ kind.

Finally, the study’s conclusions would be stronger with more data from non-English speakers, e.g., from one or two of other languages tested, using the same materials (where English becomes one of the target languages then). Given that the study was conducted on-line, it should not take very long for the authors to complete data collection.

Reviewer #2 (Remarks to the Author):

The authors examine whether foreign language words with the same meaning share more

phonological features across languages than words with different meanings, specifically whether the ability to infer meaning from form increases with the degree to which a given word conforms to cross-linguistic regularities in form-meaning mapping. They also examine whether detection and use of form-meaning regularities are moderated by individual differences in verbal working memory.

This paper is highly challenging and interesting. I want the authors to refer to some classic linguistic studies such as Jespersen (1933) and Bloomfield (1933).

O. Jespersen, "The symbolic value of the vowel i," *Philologica*, vol. 1, pp. 1-19, 1933.

L. Bloomfield, "Language," New York: Henry Holt, 1933.

There are also studies related to sound symbolism in taste.

- Maki Sakamoto and Junji Watanabe: Cross-Modal Associations between Sounds and Drink Tastes/Textures: A Study with Spontaneous Production of Sound-Symbolic Words, *Chemical Senses*, 41, 197-203.

DOI: 10.1093/chemse/bjv078 (2016)

- Crisinel, A. S., Jones, S., & Spence, C. (2012).

'The sweet taste of maluma': Crossmodal associations between tastes and words. *Chemosensory Perception*, 5, 266– 273.

- Gallace, A., Boschini, E., & Spence, C. (2011).

On the taste of "Bouba" and "Kiki": An exploration of word-food associations in neurologically normal participants.

Cognitive Neuroscience, 2, 34– 46.

doi.org/10.1080/17588928.2010.516820

Related to sound symbolism in personality, the authors may be interested in Maki Sakamoto, Junji Watanabe, Koichi Yamagata: Automatic Estimation of Multidimensional Personality From a Single Sound-Symbolic Word, *Frontiers in Psychology*, 12(595986), 1-16. DOI: 10.3389/fpsyg.2021.595986 (2021)

The authors should discuss the limitations of this study more fully. For example, they should consider how the meaning of existing words in each language can affect inference of the meaning of words.

Reviewer #3 (Remarks to the Author):

Review of "How the mind finds meaning in the sounds of language"

The study investigates the ability of speakers to guess the meaning of word pairs with antonymic meanings in foreign languages. While the study itself seems to be very interesting, and the authors seem to have tried hard to make a good case for it, also providing a good literature review (as far as I can evaluate this), I see a lack of clarity in the methods, in so far as certain measures are not clearly enough described to the readers and/or lacking in the code. I also see a problem in the way in which the "linguistic distances" were computed. I therefore recommend to revise the study and resubmit it, so that it can receive another round of reviews.

Major Remarks

The major problem I have with the study in the current form is that the experiments conducted by the authors are not very well described here. I am asking myself: where are the stimuli? Where do I find the list of antonym pairs used in all 9 languages plus English, how were the words represented, so that they could compare their phonetic distances with the English words, where do I find the code with which the distances were calculated? All of these aspects are crucial, also for replication, not only the individual results of informants guessing an item. So I would suggest that the authors revise their data and provide all stimulus data (the list of words used, with transcriptions in all 9 languages, also the code to compute phonetic distances, and the like). I also suggest that the authors illustrate their experiments more clearly, by providing a box or a figure that shows the reader what exactly was done, since this is one of the most interesting parts of the experiment. As a radical proponent of open science, I'd say, the stimuli (the synthesized words) should also be shared in the form of audio files to give interested scholars and readers the possibility to understand from what information informants had to guess.

I do not understand and think it does not make any sense to transcribe words in X-Sampa and then compute phonetic distances between them using the Levensthein distance given that there are many much better tools for the calculation of phonetic distances available now (which have already been applied in the context of research in psychology, see Gampe, Kurthen, and Daum 2017). I would strongly suggest to use clear phonetic transcriptions in IPA and then apply mature methods for the calculation of phonetic distances like the SCA method provided by LingPy (<https://lingpy.org>, see List 2014; List et al. 2018), or some alternative methods like the one proposed by Dellert (2017) or Kilani (2020). I would suggest to not call those distances "linguistic distances" but "phonetic distances", since "linguistic distances" should be reserved for the distances between languages overall (according to the mainstream literature in comparative linguistics).

Minor Remarks

- I think the authors should mention Winter et al. (2022), since this is a recent and new study on sound symbolism on a large cross-linguistic dataset.
- I welcome the use of the original orthography in scientific texts, e.g., for Japanese, accompanied by the transliteration. For languages like Japanese and Chinese, it is also easier to understand than what original characters are used to spell the respective words, given the large number of homophones in both languages.
- If you mention "Levenshtein distances", you should quote Levenshtein (1966), but see above, where I argue to drop these calculations in favor of better measures developed in the past years.

Final Remarks

I should say that I find the study quite interesting and think it may provide a good fit for the journal, so my criticisms should be taken as encouragement.

References

Dellert, Johannes. 2017. "Information-Theoretical Causal Inference of Lexical Flow." PhD, Tübingen: Eberhard-Karls Universität.

- Gampe, Anja, Ira Kurthen, and Moritz M. Daum. 2017. "BILEX: A New Tool Measuring Bilingual Children's Lexicons and Translational Equivalents." *First Language* 0 (0): 0142723717736450. <https://doi.org/10.1177/0142723717736450>.
- Kilani, Marwan. 2020. "FAAL: A Feature-Based Aligning ALgorithm." *Language Dynamics and Change* 11 (1): 30–76. <https://doi.org/10.1163/22105832-01001300>.
- Levenshtein, V. I. 1966. "Binary Codes Capable of Correcting Deletions, Insertions, and Reversals." *Soviet Physics Doklady* 10 (8): 707–10.
- List, Johann-Mattis. 2014. *Sequence Comparison in Historical Linguistics*. Düsseldorf: Düsseldorf University Press.
- List, Johann-Mattis, Mary Walworth, Simon J. Greenhill, Tiago Tresoldi, and Robert Forkel. 2018. "Sequence Comparison in Computational Historical Linguistics." *Journal of Language Evolution* 3 (2): 130–44.
- Winter, Bodo, Márton Sóskuthy, Marcus Perlman, and Mark Dingemanse. 2022. "Trilled /r/ Is Associated with Roughness, Linking Sound and Touch Across Spoken Languages." *Scientific Reports* 12 (1). <https://doi.org/10.1038/s41598-021-04311-7>.

Reviewer #1

- 1) I find the equation of cross-linguistic form-meaning regularity and sound symbolism problematic, in the absence of a discussion or argument that sound symbolism is the only or most likely account of cross-linguistic form-meaning regularity. This cannot just be assumed but has to be motivated, or least it has to be explicitly recognized that it is an assumption that may be overturned by new theoretical ideas on what could constrain form-meanings mappings to be non-arbitrary, beyond resemblance between sound and meaning.

Concretely, the authors write: “A measure of cross-linguistic form-meaning regularity (henceforth referred to as “sound symbolism”) was calculated by estimating the extent to which words with the same meaning had greater phonological overlap across words in different languages than words with opposite meanings.” But this requires showing independently (in a triangulation procedure) that there is sound-symbolism at least in one of the languages at issue: i.e., that ‘sharp’ resembles (sound-symbolically) to the meaning [sharp], or that ‘surudoï’ does, or both. The current measure is not specific enough and will capture all sorts of other non-arbitrary relations between forms across languages for words that share the same meanings: indeed, the strongest effects are found for historically related languages in this study, i.e., languages that share a common origin of large portions of the vocabulary through the common Latin root. I agree with the authors that their results indicate that “the ability to infer the meanings of foreign words cannot be fully attributed to direct cross-linguistic transfer at the lexical level”, but that does automatically validate an explanation based on sound symbolism.

We thank the Reviewer for raising this important point and have addressed it in two ways. First, we have made revisions throughout the Methods and Results to change references to the “Sound Symbolism” measure to the more appropriate label of “Form-Meaning Regularity” to avoid undue speculation regarding the cause of cross-linguistic similarities. Second, we have revised the Discussion to acknowledge this limitation, making it clear that while the present data provides evidence of consistent form-meaning regularities, further research will be needed to empirically establish iconic relations between the phonological forms and their referents (p. 11).

- 2) It would be important to know more about how the 45 English antonym word pairs were chosen in the first place, and for how many there actually was/wasn’t sound symbolism of the ‘sharp’ kind.

The revised submission includes greater detail regarding the methods, including how the 45 English antonym word pairs (now presented in Table 1) were chosen. We began by gathering an initial set of 60 antonym pairs from lists of common English words. Native speakers of the non-English languages were then consulted to remove word pairs with ambiguous, inexact, or otherwise problematic translations and select a final set of 15 noun, verb, and adjective word pairs that had clear and unambiguous translations in each of the 9 non-English languages. As a goal of the present investigation was to determine the extent to which consistent form-meaning regularities could be observed in natural language (beyond onomatopoeias), the word pairs were deliberately selected without regard to overt iconicity. As noted in the previous response (and in the Discussion of the revised submission), this approach does come with the limitation that while we have demonstrated consistent covariation between form and meaning across a variety of languages, the present data does not speak to the degree of resemblance between word forms and their physical referents.

- 3) Finally, the study's conclusions would be stronger with more data from non-English speakers, e.g., from one or two of other languages tested, using the same materials (where English becomes one of the target languages then). Given that the study was conducted on-line, it should not take very long for the authors to complete data collection.

As advised, the revised submission now includes a follow-up study with 46 native Spanish speakers tested with English, a Japonic-Sino-Tai language, and a Slavic language. As in the initial study, we observed that native Spanish speakers (with minimal foreign language experience, including with English) were able to accurately infer the meanings of foreign language words significantly better than chance. Effects of phonetic distance (both between the foreign and native words and among the foreign language words) were also remarkably consistent with the findings from the initial study. The main point of deviation was that there was a notably smaller effect of verbal working memory among native Spanish speakers; the implications and potential causes of this difference are considered on p. 11-12 of the Discussion.

Reviewer #2 (Remarks to the Author):

- 4) This paper is highly challenging and interesting. I want the authors to refer to some classic linguistic studies such as Jespersen (1933) and Bloomfield (1933). O. Jespersen, "The symbolic value of the vowel i," *Philologica*, vol. 1, pp. 1-19, 1933. L. Bloomfield,

“Language,” New York: Henry Holt, 1933. There are also studies related to sound symbolism in taste.

- Maki Sakamoto and Junji Watanabe: Cross-Modal

Associations between Sounds and Drink Tastes/Textures:
A Study with Spontaneous Production of Sound-Symbolic
Words, *Chemical Senses*, 41, 197-203.

DOI: 10.1093/chemse/bjv078 (2016)

- Crisinel, A. S., Jones, S., & Spence, C. (2012).

‘The sweet taste of maluma’: Crossmodal associations between
tastes and words. *Chemosensory Perception*, 5, 266– 273.

- Gallace, A., Boschini, E., & Spence, C. (2011).

On the taste of “Bouba” and “Kiki”: An exploration of word–
food associations in neurologically normal participants.

Cognitive Neuroscience, 2, 34– 46.

doi.org/10.1080/17588928.2010.516820

Related to sound symbolism in personality, the authors may be interested in Maki Sakamoto, Junji Watanabe, Koichi Yamagata: Automatic Estimation of Multidimensional Personality From a Single Sound-Symbolic Word, *Frontiers in Psychology*, 12(595986), 1-16. DOI: 10.3389/fpsyg.2021.595986 (2021)

We thank the Reviewer for these helpful suggestions, which have been incorporated on p. 3 to bolster claims regarding the existence of cross-modal associations between sound, taste, and personality.

- 5) The authors should discuss the limitations of this study more fully. For example, they should consider how the meaning of existing words in each language can affect inference of the meaning of words.

The revised Discussion now includes a section detailing outstanding questions regarding the cause of form-meaning regularities and the ability to infer the meanings of unfamiliar foreign words (p. 11). Specifically, we acknowledge that form-meaning regularities may emerge from causes other than linguistic iconicity and that listeners may be able to rely on form-meaning connections in their own language to infer the meanings of new words, even in the absence of iconic sound symbolism.

Reviewer #3 (Remarks to the Author):

- 6) The major problem I have with the study in the current form is that the experiments conducted by the authors are not very well described here. I am asking myself: where are the stimuli? Where do I find the list of antonym pairs used in all 9 languages plus English, how were the words represented, so that they could compare their phonetic distances with the English words, where do I find the code with which the distances were calculated? All of these aspects are crucial, also for replication, not only the individual results of informants guessing an item. So I would suggest that the authors revise their data and provide all stimulus data (the list of words used, with transcriptions in all 9 languages, also the code to compute phonetic distances, and the like). I also suggest that the authors illustrate their experiments more clearly, by providing a box or a figure that shows the reader what exactly was done, since this is one of the most interesting parts of the experiment. As a radical proponent of open science, I'd say, the stimuli (the synthesized words) should also be shared in the form of audio files to give interested scholars and readers the possibility to understand from what information informants had to guess.

As suggested, we have included the full list of English antonym word pairs in the Methods section of the revised manuscript (see Table 1). We have also included a figure depicting an overview of the procedure (Figure 7) in addition to the example of an antonym word pair trial included in the previous version (Figure 6). Finally, the revised submission now includes a link to a public repository with the R code and data used in the primary analysis, the full list of antonym pairs and their IPA transcriptions in English and the 9 non-English languages, the Python code used to calculate phonetic distances, and the audio files utilized in the experiment itself. These materials can be accessed through the following link: <https://osf.io/4ez8v/>

- 7) I do not understand and think it does not make any sense to transcribe words in X-Sampa and then compute phonetic distances between them using the Levenstein distance given that there are many much better tools for the calculation of phonetic distances available now (which have already been applied in the context of research in psychology, see Gampe, Kurthen, and Daum 2017). I would strongly suggest to use clear phonetic transcriptions in IPA and then apply mature methods for the calculation of phonetic distances like the SCA method provided by LingPy (<https://lingpy.org>, see List 2014; List et al. 2018), or some alternative methods like the one proposed by Dellert (2017) or Kilani (2020). I would suggest to not call those distances “linguistic distances” but “phonetic distances”, since “linguistic distances” should be reserved for the distances between languages overall (according to the mainstream literature in comparative linguistics).

We thank the Reviewer for this helpful suggestion and have since revised all analyses that previously utilized Levenstein distances. Specifically, phonetic distances between words were recalculated using the SCA method provided by LingPy based on IPA transcribed words. We have additionally replaced references to both “linguistic distance” and “Levenstein distance” with “phonetic distance” as advised throughout the manuscript.

Minor Remarks

- 8) I think the authors should mention Winter et al. (2022), since this is a recent and new study on sound symbolism on a large cross-linguistic dataset.

Winters et al. (2022) is indeed relevant to the present work and is now referenced in the introduction on p. 3 as follows:

A number of recent studies utilizing large cross-linguistic datasets indicate that systematic form-meaning associations can indeed be found in natural languages^{9,42}. For instance, Winter et al.⁴² demonstrated that words related to sensations of roughness are particularly likely to include a trilled /r/ sound, and that this pattern holds true across a diverse set of languages and language families.

- 9) I welcome the use of the original orthography in scientific texts, e.g., for Japanese, accompanied by the transliteration. For languages like Japanese and Chinese, it is also easier to understand then what original characters are used to spell the respective words, given the large number of homophones in both languages.

As advised, foreign language (specifically Japanese) words referenced in the text now include the original Japanese characters as well as their English transliteration.

- 10) If you mention “Levenshtein distances”, you should quote Levenshtein (1966), but see above, where I argue to drop these calculations in favor of better measures developed in the past years.

We thank the Reviewer again for their excellent suggestion regarding the use of more sophisticated measures. We have recalculated our distance measures using LingPy’s SCA method and replaced references to Levenstein distances with “phonetic distances.”

6th Sep 23

Dear Dr. Hayakawa,

Your manuscript titled "How the mind finds meaning in the sounds of language" has now been seen by our reviewers, whose comments appear below. In light of their advice I am delighted to say that we are happy, in principle, to publish a suitably revised version in Communications Psychology under the open access CC BY license (Creative Commons Attribution v4.0 International License).

We therefore invite you to revise your paper one last time to address a list of editorial requests. At the same time we ask that you edit your manuscript to comply with our format requirements and to maximise the accessibility and therefore the impact of your work.

EDITORIAL REQUESTS:

Furthermore, we think a matter-of-fact presentation that stays close to the paradigm is key to the presentation of the work.

SUBMISSION INFORMATION:

OPEN ACCESS:

Communications Psychology is a fully open access journal. Articles are made freely accessible on publication under a [CC BY license](http://creativecommons.org/licenses/by/4.0) (Creative Commons Attribution 4.0 International License). This license allows maximum dissemination and re-use of open access materials and is preferred by many research funding bodies.

For further information about article processing charges, open access funding, and advice and support from Nature Research, please visit <https://www.nature.com/commspsychol/article-processing-charges>

At acceptance, you will be provided with instructions for completing this CC BY license on behalf of all authors. This grants us the necessary permissions to publish your paper. Additionally, you will be

asked to declare that all required third party permissions have been obtained, and to provide billing information in order to pay the article-processing charge (APC).

* **DATA AVAILABILITY:**

[link redacted]

Best regards,

Jennifer Bellingtier

Jennifer Bellingtier, PhD
Senior Editor
Communications Psychology

REVIEWERS' EXPERTISE:

Reviewer #1 Psycholinguistics

Reviewer #2 Psycholinguistics

Reviewer #3 Psycholinguistics

REVIEWERS' COMMENTS:

Reviewer #1 (Remarks to the Author):

I think the authors have done a good job addressing my comments and/or acknowledging limitations of their approach.

Reviewer #2 (Remarks to the Author):

I think you have made the necessary corrections, so I have no further comments.

Reviewer #3 (Remarks to the Author):

I am very pleased with the changes introduced by the authors and have no further objections against the publication of the manuscript in this form.